



# Exploiting Aeolus Level-2B Winds to Better Characterize Atmospheric Motion Vector Bias and Uncertainty

Katherine E. Lukens[1,2], Kayo Ide[3], Kevin Garrett[1], Hui Liu[1,2], David Santek[4], Brett Hoover[4], and Ross N. Hoffman[1,2]

[1]NOAA/NESDIS/Center for Satellite Applications and Research (STAR), College Park, Maryland, 20740, USA
[2]Cooperative Institute for Satellite Earth System Studies (CISESS), University of Maryland, College Park, Maryland, 20740, USA
[3]University of Maryland, College Park, Maryland, 20740, USA
[4]Cooperative Institute for Meteorological Satellite Studies (CIMSS), University of Wisconsin-Madison, Madison, Wisconsin, 53706, USA

*Correspondence to*: Katherine E. Lukens (katherine.lukens@noaa.gov)

**Abstract.** The need for highly accurate atmospheric wind observations is a high priority in the science community, and in particular numerical weather prediction (NWP). To address this requirement, this study leverages Aeolus wind LIDAR Level-2B data provided by the European Space Agency (ESA) to better characterize atmospheric motion vector (AMV) bias and uncertainty, with the eventual goal of potentially improving AMV algorithms. AMV products from geostationary (GEO) and low-Earth polar orbiting (LEO) satellites are compared with reprocessed Aeolus horizontal line-of-sight (HLOS) global winds observed in August and September 2019. Winds from two of the four Aeolus observing modes are utilized for comparison with AMVs: Rayleigh-clear (derived from the molecular scattering signal) and Mie-cloudy (derived from particle scattering). For the most direct comparison, quality controlled (QC'd) Aeolus winds are collocated with quality controlled AMVs in space and time, and the AMVs are projected onto the Aeolus HLOS direction. Mean collocation differences (MCD) and standard deviation (SD) of those differences (SDCD) are determined from comparisons based on a number of conditions, and their relation to known AMV bias and uncertainty estimates is discussed. GOES-16 and LEO AMV characterizations based on Aeolus winds are described in more detail.

Overall, QC'd AMVs correspond well with QC'd Aeolus HLOS wind velocities (HLOSV) for both Rayleigh-clear and Mie-cloudy observing modes, despite remaining biases in Aeolus winds after reprocessing. Comparisons with Aeolus HLOSV are consistent with known AMV bias and uncertainty in the tropics, NH extratropics, and in the Arctic, and at mid- to upper-levels in both clear and cloudy scenes. SH comparisons generally exhibit larger than expected SDCD, which could be attributed to height assignment errors in regions of high winds and enhanced vertical wind shear. GOES-16 water vapor clear-sky AMVs perform best relative to Rayleigh-clear winds, with small MCD (-0.6 m s$^{-1}$ to 0.1 m s$^{-1}$) and SDCD (5.4-5.6 m s$^{-1}$) in the NH and tropics that fall within the accepted range of AMV error values relative to radiosonde winds. Compared to Mie-cloudy winds, AMVs exhibit similar MCD and smaller SDCD (~4.4-4.8 m s$^{-1}$) throughout the troposphere. In polar regions, Mie-



cloudy comparisons have smaller SDCD (5.2 m s$^{-1}$ in the Arctic, 6.7 m s$^{-1}$ in the Antarctic) relative to Rayleigh-clear comparisons, which are larger by 1-2 m s$^{-1}$.

The level of agreement between AMVs and Aeolus winds varies per combination of conditions including the Aeolus observing mode coupled with AMV derivation method, geographic region, and height of the collocated winds. It is advised that these stratifications be considered in future comparison studies and impact assessments involving 3D winds. Additional bias corrections to the Aeolus dataset are anticipated to further refine the results.

# 1 Introduction

The need to improve atmospheric 3D wind observations in the troposphere has long been a high priority in the science community. In 2018, the National Academies Press published the 2017-2027 decadal survey for Earth science and applications from space (National Academies, 2018), which includes 3D winds in a series of observation requirement priorities and accompanying recommendations. The survey recommends that radiometry-based atmospheric motion vector (AMV) tracking should be able to address the priority requirement of 3D winds.

AMVs are wind observations derived from tracking clouds and water vapor features in satellite images through time. Both geostationary (GEO) and polar-orbiting, i.e., low-earth orbiting (LEO), satellites observe the motion of such features in several spectral regions. Infrared bands that are specifically sensitive to water vapor (WV) absorption capture atmospheric motion in two ways: (1) water vapor cloud-top (WVcloud) channels are used to track upper-level cloud top motions, and (2) water vapor clear-sky (WVclear) channels are used to detect upper-tropospheric features (e.g., jet stream and waves) by tracking water

vapor motions in clear air (Velden et al., 1997). Infrared window (hereafter IR) cloud-track AMVs are based on longwave and shortwave channels that are useful for detecting motions in cloudy scenes at mid- to upper-levels (e.g., cirrus clouds) and lower levels (e.g., low stratus clouds and fog), respectively (Velden et al., 2005).

AMVs are regularly assimilated in numerical weather prediction (NWP), and they have been shown to positively impact operational forecast skill (e.g., Le Marshall et al., 2008; Berger et al., 2011; Wu et al., 2014). Since NWP data assimilation

(DA) methods assume knowledge of observational error statistics, any improved characterization of AMV observation errors has the potential to improve NWP DA and hence forecast skill.

Until recently, AMVs were one of a few sources of vertically varying 2D wind observations. Aeolus is a novel polar-orbiting satellite that was launched in 2018 by the European Space Agency (ESA) to observe vertical wind profiles from space (Stoffelen et al., 2005; ESA, 2008; Straume-Lindner, 2018). Onboard Aeolus is a Doppler Wind Lidar (DWL) instrument

(Reitebuch et al., 2009) which observes winds converted from backscatter retrievals along the line-of-sight (LOS) of the DWL laser. Rayleigh and Mie receivers detect molecular backscattering and aerosol and cloud backscattering, respectively (Straume et al., 2018) and are converted into horizontal LOS (HLOS) wind velocities (HLOSV). Rayleigh and Mie receivers observe both clear and cloudy scenes; hence, the resultant wind retrievals fall into one of four possible observing modes: Rayleigh-clear, Rayleigh-cloudy, Mie-clear, and Mie-cloudy. Rayleigh-clear and Mie-cloudy winds are of better quality and are



recommended for use in analysis based on NWP assessments by ESA and ECMWF (Rennie and Isaksen, 2019; Rennie et al., 2020). Rayleigh-cloudy winds are not typically used as they sample the same locations as Mie-cloudy winds and are generally contaminated by the Mie channel. Mie-clear winds are routinely discarded as they are of poorer quality since the Mie backscattered signal is dominated by noise in clear conditions (Rennie et al., 2020; Abdalla et al., 2020).

This study aims to leverage Aeolus Level-2B (L2B) HLOS wind profiles as a standard for comparison to characterize AMV
observation bias and uncertainty, with the eventual goal of potentially improving AMV algorithms and the impact of AMV observations on NWP skill. The availability of the Aeolus dataset provides the unique opportunity to directly assess the performance of AMVs derived from different retrieval channels relative to a global reference wind profile dataset observed by a single unit. Such a direct comparison has not previously been possible due to the sparse spatial coverage of other available reference datasets, e.g., rawinsonde winds (e.g., Chen et al., 2021; Liu, B. et al., 2021; Martin et al., 2021). Further, Aeolus
observations are made at a set of fixed vertical levels that represent the averages of accumulated measurements within vertical range bins. The thickness of these range bins increases with height to mitigate the decrease in signal strength with height (Rennie and Isaksen, 2020a). As such, height-related HLOS wind errors should be small relative to errors in AMV height assignment.

The structure of the paper is as follows: Section 2 describes the datasets used. Section 3 defines the quality controls, collocation
methodology, and skill metrics. Section 4 assesses the overall performance of AMVs with respect to collocated Aeolus wind observations, and discusses the characterization of AMV mean collocation difference (MCD) and standard deviation (SD) of collocation differences (SDCD) based on different sets of conditions. AMV performance metrics specific to GOES-16 and the suite of available LEO satellites are described in more detail. Section 5 summarizes the findings.

## 2 Data

### 2.1 Aeolus Level-2B winds

Aeolus Level-2B wind profiles (de Kloe, 2019; de Kloe et al., 2020) used in this study are derived from retrievals from the satellite's backup laser, known as Flight Model-B (FM-B), that was switched on in 2019. The L2B wind product consists of geo-located vector wind profiles projected along the HLOS of the FM-B laser, which points away from the sun (i.e., perpendicular to the spacecraft track) at 35° off nadir. Because of the terminator orbit and sensor geometry away from the
poles, winds in ascending orbits (southeast to northwest direction) are observed around sunset (local equator crossing time is 18:00 UTC), and winds in descending orbits (northeast to southwest direction) are observed around sunrise (local equator crossing time is 06:00 UTC). The satellite completes one orbit around Earth in approximately 92 minutes, and 7 days is the repeat cycle.

This study uses Aeolus wind profiles (baseline B10 product) during the period of 2 August – 16 September 2019. The selected
period of study was recommended by ESA for analysis as the Aeolus data are more stable and biases are relatively small (Rennie and Isaksen, 2019; Rennie and Isaksen, 2020a). The Aeolus winds were reprocessed by ESA using the updated L2B



processor v3.3 that includes the M1 mirror temperature bias correction that was activated on 20 April 2020 (Rennie and Isaksen, 2020a). The M1 mirror temperatures are scene-dependent and vary based on the top-of-atmosphere radiation. Specifically, the M1 mirror reflects and focuses the backscattered laser signal onto the Rayleigh and Mie receivers. Therefore,

changes in the mirror shape due to thermal variations result in perceived frequency shifts of the signal. The operational M1 bias correction uses instrument temperatures as predictors and innovation departures from ECMWF backgrounds as a reference, and is shown to improve the quality of the Rayleigh and Mie signal levels, reducing the Aeolus HLOS wind bias relative to ECMWF background winds by over 80%: the global average Rayleigh-clear bias decreased to near-zero and the Mie bias decreased to -0.15 m s$^{-1}$ (Abdalla et al., 2020; information regarding the limitations of the operational M1 correction

are presented in Weiler et al., 2021). In this study, profiles of Aeolus Rayleigh-clear HLOS winds (hereafter RAY winds) and Mie-cloudy HLOS winds (hereafter MIE winds) are collocated with AMVs. The AMVs projected onto the collocated Aeolus HLOS will be referred to as AMV winds and the original AMVs will be referred to as AMV wind vectors hereafter. Data from the other observing modes (Rayleigh-cloudy and Mie-clear) are of poorer quality and quantity and are not used.

ECMWF conducted several studies to verify the quality of Aeolus observations (e.g., Rennie and Isaksen, 2019; de Kloe et al.,

2020). They found that with the application of the M1 bias correction and proper quality controls (QC) as well as Aeolus black-listed dates taken into account, Aeolus provides high quality wind observations relative to ECMWF background. RAY winds minus ECMWF IFS HLOS winds have a global mean of -0.04 m s$^{-1}$ and a standard deviation of 5.3 m s$^{-1}$. MIE minus IFS winds have a global mean of -0.16 m s$^{-1}$ and a smaller standard deviation of 3.8 m s$^{-1}$ (Abdalla et al., 2020). It is noted that the ECMWF model, the Integrated Forecasting System (IFS), is used as a reference in the calculation of the reprocessed Aeolus

L2B winds, and thus a model dependency is introduced into the dataset (Weiler et al., 2021). Related NWP impact assessments show that Aeolus has a positive impact on operational global forecasts (Cress 2020; Rennie and Isaksen 2020b) at major NWP centers including ECMWF, the German Weather Service (DWD), Météo-France, and the UK Met Office. Additionally, recent studies have compared Aeolus winds with various benchmark wind datasets (e.g., rawinsondes and reanalyses). For example, Santek et al. (2021) found that when taking collocated polar rawinsonde winds as the truth, quality controlled reprocessed

RAY winds share similar observation error standard deviations (5-6 m s$^{-1}$) but exhibit a larger wind speed bias of -1.1 m s$^{-1}$ with respect to comparisons of good-quality water vapor wind retrievals from the National Aeronautics and Space Administration (NASA) Aqua satellite (bias of -0.2 m s$^{-1}$). In a similar comparison to ECMWF Reanalysis v5 (ERA5), Santek et al. (2021) also found that Aeolus and Aqua share similar smaller biases (0.02-0.17 m s$^{-1}$) and uncertainties (~4.5 m s$^{-1}$) throughout the vertical.

Despite the high quality and positive impacts, limitations remain with the Aeolus L2B dataset (Abdalla et al., 2020; Weiler et al., 2021). Mie and Rayleigh random errors could be further improved, as the Mie error standard deviations average to approximately 3.5 m s$^{-1}$ and Rayleigh error standard deviations increase from 4 m s$^{-1}$ to over 5 m s$^{-1}$ from July to December 2019 (Abdalla et al., 2020). Further, MIE winds exhibit a slow (fast) wind speed dependent bias for high HLOS speeds of negative (positive) sign. Moreover, there is currently an ECMWF model dependency in the reprocessed Aeolus L2B wind

dataset (Weiler et al., 2021). Additionally, at the time of writing, issues thought to be due to instrumentation or software





malfunctions have become apparent that affect the quality of the winds. One specific issue is the decrease in the internal and atmospheric return signals that is linked to slowly increasing random errors for Rayleigh-clear winds (Straume et al., 2021). Efforts at ESA are currently underway to resolve these issues.

## 2.2 Atmospheric motion vectors

AMVs examined in this study are used in the National Oceanic and Atmospheric Administration (NOAA) National Centers for Environmental Prediction (NCEP) operations and are archived in 6-hour satellite wind (SATWND) BUFR files centered on the analysis times 00, 06, 12, and 18 UTC. AMVs derived from sequences of GEO satellite images are observed equatorward of ~60° latitude. Polar AMVs (observed at latitudes poleward of 60°) are derived in areas covered by three consecutive LEO satellite images. The GEO satellites include GOES-15 and GOES-16 operated by NOAA, Meteosat-8 and

Meteosat-11 (the first and fourth satellites in the Meteosat Second Generation (MSG) series at the European Organization for the Exploitation of Meteorological Satellites (EUMETSAT)), Himawari-8 managed by the Japan Meteorological Agency (JMA), and INSAT-3D from the Indian Space Research Organization (ISRO). GEO AMVs in this study are derived from IR, WVcloud, and WVclear channels from the GOES Imager onboard GOES-15, the Advanced Baseline Imager (ABI) onboard GOES-16, the Spinning Enhanced Visible and Infrared Imager (SEVIRI) onboard Meteosat-8 and Meteosat-11, the Advanced

Himawari Imager (AHI) onboard Himawari-8, and the INSAT Imager onboard INSAT-3D. (It is noted that Himawari-8 and INSAT-3D WVclear AMVs are not included in the NCEP data archive.)

AMVs from LEO satellites include several operated by NOAA: NOAA-15, -18, -19, -20 and Suomi National Polar-orbiting Partnership (S-NPP). Additional LEO satellites include MetOp-A and MetOp-B operated by EUMETSAT, and Aqua and Terra operated by NASA. LEO AMVs considered for analysis are derived from cloud-track IR window channels from instruments

including but not limited to: The Visible and Infrared Imaging Radiometer Suite (VIIRS) onboard NOAA-20 and S-NPP; the Advanced Very High Resolution Radiometer (AVHRR) instrument onboard NOAA-15, -18, -19, MetOp-A, and MetOp-B; and the Moderate Resolution Imaging Spectroradiometer (MODIS) onboard Aqua and Terra.

Numerous studies have evaluated bias and uncertainty characteristics of AMVs through direct comparison with in situ radio-/rawinsonde observations and NWP analyses (e.g., Velden et al., 1997; Bormann et al., 2002, 2003; Le Marshall et al., 2008;

Bedka et al., 2009; Velden and Bedka, 2009; Key et al., 2016; Daniels et al., 2018; Cotton et al., 2020). The derived motion wind algorithms that generate AMVs from IR, WVcloud, and WVclear channels can vary between centers (Santek et al., 2014; Santek et al., 2019). Available AMV performance metrics are presented in Table 1 and include results from comparisons of all GEO AMVs and from specific examples of GEO and LEO satellites (GOES-16, and Aqua and Terra, respectively).



**Table 1: Summary of published statistics of AMV performance. IR indicates the IR-window channel. NA denotes unavailable information from the sources used. Sources include [1]Santek et al. (2019) for July, AMV QI ≥ 80%; [2]Daniels et al. (2018) for November, QI ≥ 60%; [3]Cotton et al. (2020) for November, QI ≥ 80%; [4]Key et al. (2016) for March-August, No QI used; [5]Le Marshall et al. (2008) for May-January, QI ≥ 85%.**

| AMV Source | Region | Verification | Speed Bias (m s⁻¹) | Speed SD, RMS (m s⁻¹) | Vector Diff (m s⁻¹) | Vector RMS (m s⁻¹) |
|---|---|---|---|---|---|---|
| [1]All GEO | Global | Radiosonde | -1.79 to 0.31 | 4.07 to 6.54 | NA | 5.93 to 8.97 |
| [1]All GEO | Global | NWP Analysis | -1.12 to 0.26 | 1.11 to 4.57 | 2.59 to 5.54 | 3.10 to 7.53 |
| [2]GOES-16, IR | Full Disk | Radiosonde | ~ -1.0 to 0.50 | NA | ~ 3.00 to 6.00 | NA |
| [3]GOES-16, IR | NH, upper levels | GFS Background | -0.56 | 3.54 | NA | NA |
| [3]GOES-16, IR | Tropics, upper levels | GFS Background | -0.67 | 3.61 | NA | NA |
| [3]GOES-16, IR | SH, upper levels | GFS Background | -0.06 | 3.51 | NA | NA |
| [4,5]AQUA and TERRA, IR | Poles, upper levels | Radiosonde | -0.80 to -0.50 | NA | 4.71 to 4.81 | 5.22 to 5.55 |
| [4,5]AQUA and TERRA, IR | Poles, middle levels | Radiosonde | -1.01 to -0.35 | NA | 4.20 to 4.38 | 4.79 to 5.35 |
| [4,5]AQUA and TERRA, IR | Poles, low levels | Radiosonde | -0.91 to -0.03 | NA | 3.58 to 3.92 | 4.02 to 4.82 |

AMVs have state-dependent errors that can vary based on wind speed and water vapor content and gradient (Posselt et al., 2019). Past reports show that AMVs tend to exhibit a slow speed bias (1-5 m s⁻¹) at high levels (above 400 hPa) in the extratropics and a fast bias (1-3 m s⁻¹) at middle levels (400-700 hPa) in the tropics (Bormann et al., 2002; Schmetz et al., 1993; von Bremen, 2008). Recent improvements to AMV derivation schemes, e.g., in GOES-16/17 and Himawari-8, have reduced the fast speed bias, with the residual bias largely attributed to height assignment errors (Cotton et al., 2020). Height assignments to the AMVs via satellite- and ground-based techniques (Jung et al., 2010; Salonen et al., 2015) have been shown to account for a large source of AMV uncertainty (Velden and Bedka, 2009). One factor of height assignment error is that AMVs are generally assigned to discrete levels when instead they better correlate with atmospheric motions in layers of varying depth that depend on the vertical moisture profile (Velden et al., 2005; Velden and Bedka, 2009). Moreover, speed biases and uncertainties tend to be higher at heights and in regions with strong wind shear (Bormann et al., 2002; Cordoba et al., 2017).



## 3 Approach and quality controls

Aeolus HLOS global wind profiles are collocated with satellite-derived AMVs. The collocation approach implemented here is the same employed by Hoffman et al. (2021) and follows that employed at UW-Madison/Cooperative Institute for Meteorological Satellite Studies (CIMSS) (Santek et al., 2021). AMV observations are compared with Aeolus observations from the same and neighboring 6-h cycles to account for all possible collocations. An Aeolus observation is retained for comparison with an AMV if the Aeolus observation satisfies *all* the following collocation criteria:

1. Aeolus time falls within 60 minutes of the AMV time.

2. Aeolus pressure is within $0.04 \log_{10}$(pressure) of the AMV height assignment. (Note that the log of pressures is used to account for the non-linear decrease of pressure with increasing altitude.)

3. Aeolus observation location is within 100 km horizontal great circle distance of the AMV location.

If multiple Aeolus observations satisfy these criteria for the same AMV observation, the Aeolus observation closest in distance
is retained. Then, if multiple Aeolus observations still meet all collocation criteria, the observation closest in pressure to the AMV observation is kept for analysis. Aeolus observations are collected as a line of profiles to the right of the satellite track. Since it takes approximately 92 minutes for Aeolus to complete an orbit around Earth, all observations in any one orbit that might exist within the 100 km great circle radius around an AMV observation would occur within 30 seconds of Aeolus passing overhead. The 30-second interval is irrelevant compared to the 1-hour collocation time difference criterion. Further, the only
way for Aeolus observations from two distinct orbits to be collocated with the same AMV is for the time differences of both observations relative to the AMV to be greater than 30 minutes. Therefore, we only consider the closest profile and then to select the observation from that profile closest in the vertical to the AMV. After collocation, the AMV wind vector is projected onto the HLOS direction of its paired Aeolus observation.

Once collocated, Aeolus winds and AMVs are filtered by additional QC tests to retain pairs of quality controlled (QC'd)
observations. (QC was implemented after collocation in order to test and compare the use of different QC criteria without having to repeat the collocation process.) Aeolus QC criteria were chosen following ESA's recommendations for the RAY and MIE observing modes, and these are consistent with those listed in Rennie and Isaksen (2020a). Specifically, RAY winds are rejected if winds are close to topography (pressure > 800 hPa), horizontal accumulation length < 60 km, vertical accumulation length < 0.3 km, L2B uncertainty > 12 m s$^{-1}$ at upper levels (pressure < 200 hPa), or L2B uncertainty > 8.5 m s$^{-1}$
at lower levels (pressure > 200 hPa). Similarly, MIE winds are rejected if winds are near topography (pressure > 800 hPa) or L2B uncertainty > 5 m s$^{-1}$. For AMVs, a quality indicator (QI) of at least 80% is used to filter and retain the high-quality data; this threshold is recommended for AMV studies and in NWP by the user community and has been shown to improve statistical agreement between AMV-producing centers (Santek et al., 2019).

As a case study, we examine in greater detail the performance of AMVs from GOES-16, a GEO satellite. GOES-16 AMVs
are derived from full disk images centered at 75.2° W longitude from the onboard ABI. GOES-16 cloud-top AMVs are generally of good quality and when validated against radiosonde winds exhibit a relatively small mean difference in wind



speed ranging from -1.0 m s$^{-1}$ to +0.5 m s$^{-1}$ and mean vector differences of 3-6 m s$^{-1}$ that tend to increase with height (Table 1). Figure 1 presents the GOES-16/Aeolus collocation number densities (i.e., the total number of collocated observation pairs within each grid cell on a 1.25° (~140 km) resolution map) covering the period of study. QC'd GOES-16 AMVs collocated

with QC'd RAY and MIE winds are shown in Fig. 1a and Fig. 1b, respectively. MIE collocations exhibit three bands of high-density winds along the intertropical convergence zone (ITCZ) and extratropical storm tracks, with few winds found between 0-30° S. A similar but smoother version of the MIE distributions is shown for collocated RAY winds. The MIE collocation number density is greater than that for RAY, as AMV observation density tends to be higher in very cloudy or very moist scenes (Velden et al., 1997).

For the LEO perspective, we choose to examine the performance of all LEO AMVs derived from IR window channels rather than from a single satellite. Figure 2 depicts observation number densities of QC'd LEO AMVs collocated with QC'd RAY and MIE winds in the NH and SH polar regions bounded by 60° latitude: NH RAY (Fig. 2a), SH RAY (Fig. 2b), NH MIE (Fig. 2c), and SH MIE (Fig. 2d). In general, more LEO-MIE collocation pairs pass QC and are retained in the analysis than for RAY winds. Collocations in the Arctic are found across the high latitudes with MIE comparisons exhibiting higher

concentrations poleward of Eurasia and North America. Antarctic collocations are primarily found over the western half of the continent. In this region, water vapor features are more suitable for tracking and deriving AMVs as they exist downstream of intense upper-level storm tracks (Hoskins and Hodges, 2005) in an area of higher annual precipitation (Grieger et al., 2015).

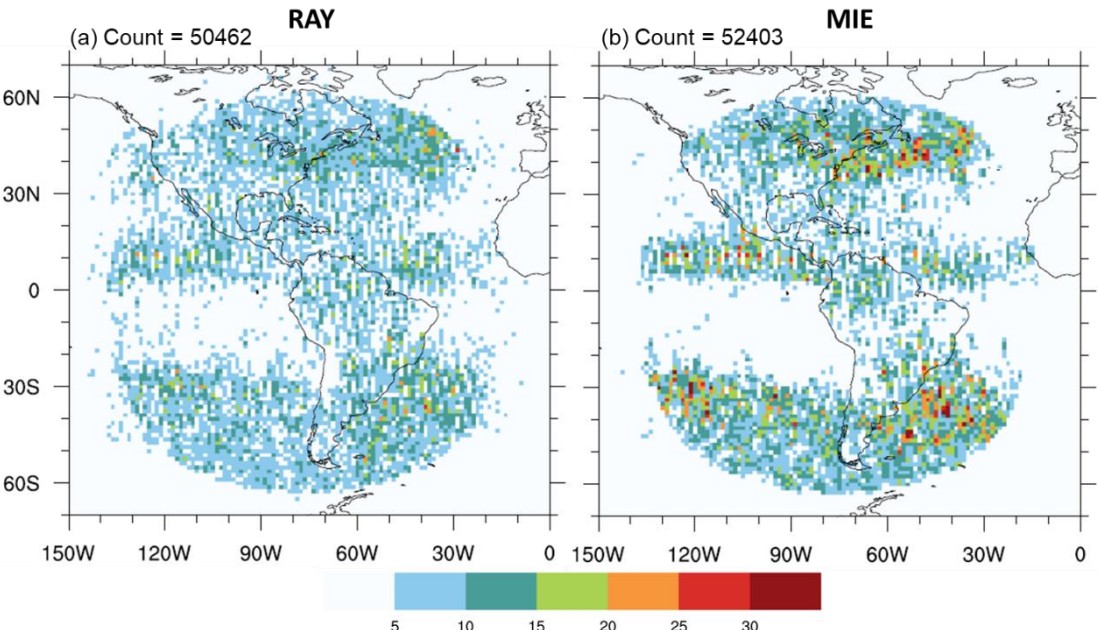

**Figure 1: Number densities of quality-controlled GOES-16 AMV observations collocated with quality-controlled Aeolus (a) Rayleigh-clear (RAY), and (b) Mie-cloudy (MIE) HLOS winds. Colors indicate total number of collocated observation pairs within a grid cell at 1.25° (140 km) horizontal resolution. Total observation count per panel is displayed in the top left corner. This and all subsequent plots are for all collocations with quality controlled AMV and Aeolus winds during the study period (2 August 2019 to 16 September 2019).**

**(a) Count = 9935**

**MIE**

**(c) Count = 19835**

**(b) Count = 11764**

**(d) Count = 16271**

1 2 3 4 5 6 7 8 9 10

**Figure 2: Number densities of IR-derived AMVs from all available LEO satellites collocated with Aeolus RAY (left column) and MIE (right column) winds in the (a,c) Arctic (north of 60° N), and (b,d) Antarctic (south of 60° S). Colors indicate total number of collocated observation pairs within a grid cell at 1.25° (140 km) horizontal resolution. Dashed latitude lines are spaced every 5 degrees. Total observation count per panel is displayed in the top left corner.**

The performance of QC'd AMVs relative to collocated Aeolus winds of QC'd are characterized by analyzing the statistics of the difference AMV minus Aeolus HLOSV. The two key statistics calculated for the collocation difference (always in the

sense of AMV minus Aeolus) are the MCD and the SDCD. It is important to emphasize that the collocation differences have several components that include errors in both AMVs and Aeolus winds. Specifically, these are due to the observation error



of the AMVs and Aeolus HLOSV, representativeness errors due to differences in scales observed, which are related to different shapes of the observing volumes, and to collocation errors due to the space and time mismatches between the observations. As previously mentioned, the estimated SD for Aeolus L2B winds is 3.8 m s$^{-1}$ for MIE and 5.3 m s$^{-1}$ for RAY, and known AMV 230 SD are shown in Table 1. In addition, differences due to collocation (i.e., due to different times and locations of the two observations) could play a role in increasing the differences between the collocated HLOS winds.

The geometry of the Aeolus observation affects how the HLOS winds are interpreted for analysis (Straume et al., 2018). The observed HLOS wind provides both a speed and direction and represents the motion of air projected onto the line-of-sight of the laser that in 2D space is nearly orthogonal to the satellite orbit direction (see Fig. 1 in de Kloe, 2019). Thus, in ascending 235 orbit segments away from the poles, a positive HLOSV indicates a westerly wind, and a negative HLOSV indicates an easterly wind; the opposite is true for winds in descending orbit segments. Figure 3 illustrates this zonal wind approximation in the tropics. The mean AMV and Aeolus HLOSV and their differences exhibit similar magnitudes of opposite sign throughout the vertical between ascending (Fig. 3a-b) and descending (Fig. 3d-e) orbit segments. This indicates that mean HLOSV differences that include winds from both ascending and descending orbit segments would be small and would represent differences of 240 larger opposing magnitudes. Moreover, the Aeolus L2B uncertainties (short dashed lines) are of similar magnitude between the orbit segments, implying that the quality of Aeolus winds is not wholly dependent on orbit segment. To simplify the interpretation of the observed HLOS winds, we multiply HLOSV in descending orbit segments by -1. In doing so, positive HLOSV (away from the poles) now indicates a westerly wind and negative HLOSV an easterly wind, regardless of Aeolus orbit segment. All statistics are based on collocation differences that include the combined ascending and minus descending orbit winds. Corresponding statistics are presented in Tables 2-5 and in Fig. 4-9.

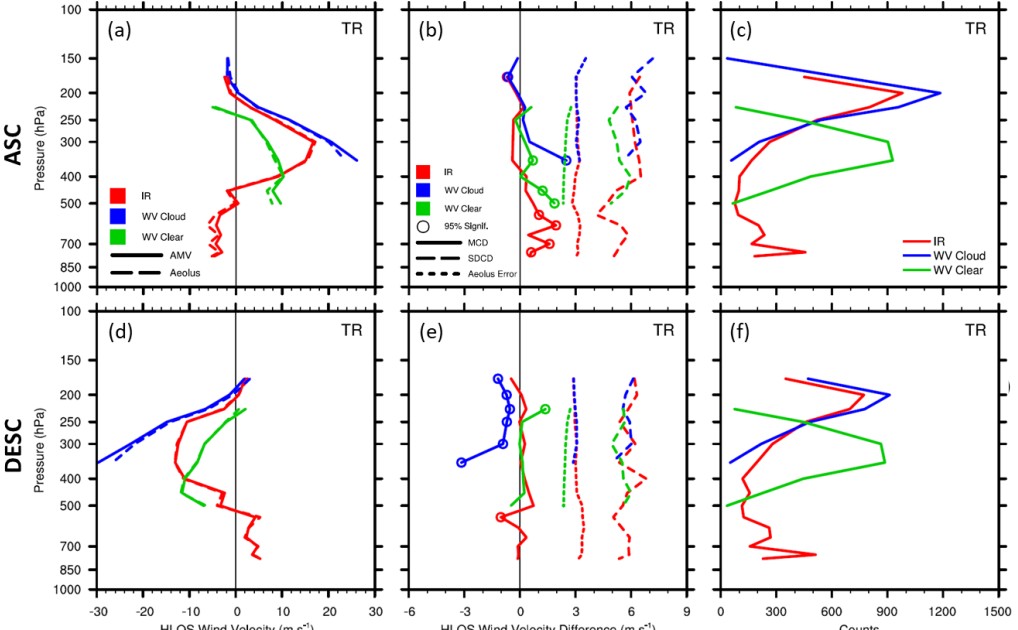

**Figure 3: Vertical comparisons of collocated GOES-16 AMVs and RAY winds in the tropics (30° S to 30° N). The top row shows Aeolus ascending orbits, (a) mean AMV (solid lines) and RAY (dashed lines) winds (m s$^{-1}$), (b) MCD (solid), SDCD (long dashed), and Aeolus L2B uncertainty (short dashed) (m s$^{-1}$), and (c) collocation counts. (d-f) as in (a-c) but for Aeolus descending orbits. Colors denote AMV channel type: IR (red), WV cloud (blue), and WV clear (green). Colored open circles indicate levels where MCD are statistically significant at the 95% level (p-value < 0.05) using the paired Student's t-test. Vertical zero lines are displayed in the left and center panels in black. Levels with observation counts > 25 are plotted.**



## 4 AMV-Aeolus comparison results

In this section we examine in detail the performance of AMVs from the GOES-16 GEO satellite and summarize the AMV performance of all LEO satellites available in the study period. We found that the level of agreement between AMVs and Aeolus winds varies per combination of conditions including the observing scene type (clear vs. cloudy) coupled with AMV derivation method, geographic region, and height of the observable.

Tables 2 and 3 summarize the performance of all available AMV HLOS winds from GEO satellites relative to Aeolus RAY and MIE winds, respectively, in the period of study; likewise, Tables 4 and 5 summarize LEO AMV performance. The statistics include correlation (r), MCD, SDCD, and root mean squared differences (RMSD) (Wilks, 2011). The correlation between collocated HLOS winds describes the overall relation of AMVs to Aeolus. The other statistics have their usual meaning applied to the HLOS wind velocities. Since the MCD are small compared to the SDCD, the RMSD and SDCD are very similar and in the following we will only discuss the SDCD, but any statement concerning the SDCD also applies to the RMSD. Using the paired two-tailed Student's $t$-test, mean differences significantly different from zero at the 90% (p-value < 0.10) and 95% (p-value < 0.05) confidence levels are indicated by bolded and underlined statistics groupings, respectively. Observation counts are also provided in the tables.

Overall, GEO AMVs correspond very well with RAY and MIE winds, with GOES and Himawari-8 AMVs having the highest correlations with Aeolus (> 0.90). MCD are small and fall within the range of known global GEO wind speed biases (Table 1). The differences vary depending on the AMV satellite, but are generally smallest in the tropics and largest in the SH extratropics where the SDCD are generally larger by ~1.0 m s$^{-1}$ relative to other regions. For GOES and Himawari-8 relative to RAY winds, the SDCD range from 5.28 m s$^{-1}$ in the NH extratropics to 6.5 m s$^{-1}$ in the SH extratropics and fall within the range of speed RMSD relative to radiosonde winds (see Table 1). Compared to other AMVs, Meteosat wind correlations are lower and corresponding SDCD values are higher by at least 2-3 m s$^{-1}$.

Performance statistics for LEO AMVs are displayed in Table 4 and Table 5. LEO AMV-Aeolus collocation pairs tend to be found at high latitudes greater than 60° and exhibit statistically significant MCD that fall within range of observed speed biases for Aqua and Terra AMVs (see Table 1). Relative to GEO, LEO AMVs exhibit higher SDCD values of ~1-2 m s$^{-1}$ for pairs with significant mean differences. In the Antarctic significant mean HLOSV differences have high SDCD values on the order of 7.5-8.5 m s$^{-1}$ for RAY and 6-7.5 m s$^{-1}$ for MIE. MIE comparisons in all regions exhibit higher correlations and lower SDCD values relative to RAY comparisons, reflecting the general higher accuracy of MIE vs. RAY winds. Another possible reason is that MIE comparisons might generally have smaller collocation errors: Aeolus MIE winds represent cloudy scenes, and IR window and WVcloud AMVs require clouds to track motions. As Aeolus RAY winds represent clear-sky and must be cloud-free, the mean great circle collocation distances tend to be slightly larger for RAY than MIE (59.5 km and 51.2 km, respectively).



**Table 2: Summary of performance characteristics of all collocations of quality-controlled GEO AMVs and quality-controlled Aeolus Rayleigh-clear (RAY) winds during the study period (2 August 2019 to 16 September 2019). Statistics include correlation (r), mean collocation differences (MCD) in m s⁻¹, standard deviation of collocation differences (SDCD) in m s⁻¹, root mean squared difference (RMSD) in m s⁻¹, and collocation count. Statistics are based on AMV minus Aeolus differences for all available AMV channel types for each Aeolus mode and at all vertical levels, and are stratified by GEO satellite and by geographic region (defined by boundaries at 30° S and 30° N), with NH indicating the Northern Hemisphere extratropics, TR the tropics, and SH the Southern Hemisphere extratropics. Bolded and bolded/underlined statistics denote collocation groupings with statistically significant mean HLOSV differences at the 90% level (p-value < 0.10) and the 95% level (p-value < 0.05), respectively. A two-sided single-variance Student's $t$-test is used to ascertain statistical significance based on the null hypothesis that the two sample populations have the same mean.**

| Geographic Region | GEO Satellite | r | MCD (m s⁻¹) | SDCD (m s⁻¹) | RMSD (m s⁻¹) | Count |
|---|---|---|---|---|---|---|
| NH | GOES-15 | **0.93** | **-0.29** | **5.40** | **5.40** | **10823** |
| NH | GOES-16 | **0.90** | **-0.35** | **5.86** | **5.87** | **13912** |
| NH | Himawari-8 | **0.93** | **0.13** | **5.28** | **5.28** | **8048** |
| NH | INSAT 3D | **0.91** | **-0.98** | **5.41** | **5.50** | **934** |
| NH | Meteosat-8 | **0.81** | **-0.26** | **7.52** | **7.53** | **8302** |
| NH | Meteosat-11 | 0.86 | 0.04 | 7.31 | 7.31 | 9234 |
| TR | GOES-15 | **0.95** | **-0.17** | **5.46** | **5.46** | **15577** |
| TR | GOES-16 | **0.94** | **0.17** | **5.82** | **5.83** | **22478** |
| TR | Himawari-8 | **0.93** | **0.22** | **5.70** | **5.71** | **28494** |
| TR | INSAT 3D | **0.92** | 0.07 | 5.80 | 5.80 | 1588 |
| TR | Meteosat-8 | **0.75** | **-2.03** | **9.22** | **9.44** | **18934** |
| TR | Meteosat-11 | 0.84 | -0.07 | 8.58 | 8.58 | 16198 |
| SH | GOES-15 | **0.94** | **-1.04** | **5.76** | **5.85** | **3949** |
| SH | GOES-16 | **0.92** | **-0.55** | **6.51** | **6.54** | **14072** |
| SH | Himawari-8 | **0.92** | **-0.92** | **5.56** | **5.63** | **7906** |
| SH | INSAT 3D | **0.95** | **-1.10** | **6.13** | **6.22** | **784** |
| SH | Meteosat-8 | **0.82** | **-1.48** | **9.26** | **9.38** | **9817** |
| SH | Meteosat-11 | **0.82** | **-0.57** | **10.18** | **10.20** | **9705** |






**Table 3: As in Table 2, but for Aeolus Mie-cloudy (MIE) winds.**

| Geographic Region | GEO Satellite | r | MCD (m s⁻¹) | SDCD (m s⁻¹) | RMSD (m s⁻¹) | Count |
|---|---|---|---|---|---|---|
| NH | GOES-15 | **0.95** | **-0.49** | **4.63** | **4.66** | **10032** |
| NH | GOES-16 | **0.94** | **-0.67** | **4.72** | **4.77** | **14301** |
| NH | Himawari-8 | **0.95** | **-0.42** | **4.54** | **4.56** | **8237** |
| NH | INSAT 3D | 0.90 | -0.01 | 5.32 | 5.32 | 476 |
| NH | Meteosat-8 | **0.91** | **-1.05** | **5.66** | **5.76** | **7143** |
| NH | Meteosat-11 | **0.93** | **-1.27** | **5.66** | **5.80** | **7719** |
| TR | GOES-15 | **0.97** | **-0.43** | **4.29** | **4.31** | **13575** |
| TR | GOES-16 | **0.97** | **-0.32** | **4.68** | **4.69** | **19945** |
| TR | Himawari-8 | 0.96 | -0.04 | 4.41 | 4.41 | 28511 |
| TR | INSAT 3D | **0.86** | **-0.52** | **5.53** | **5.55** | **796** |
| TR | Meteosat-8 | **0.86** | **0.14** | **6.14** | **6.14** | **18976** |
| TR | Meteosat-11 | **0.94** | **-0.31** | **5.45** | **5.46** | **12262** |
| SH | GOES-15 | **0.95** | **-0.79** | **5.38** | **5.44** | **4097** |
| SH | GOES-16 | **0.96** | **-1.42** | **5.05** | **5.25** | **18157** |
| SH | Himawari-8 | **0.95** | **-1.03** | **4.79** | **4.90** | **6798** |
| SH | INSAT 3D | **0.87** | **-0.88** | **5.73** | **5.79** | **438** |
| SH | Meteosat-8 | **0.93** | **-2.19** | **6.34** | **6.71** | **10699** |
| SH | Meteosat-11 | **0.92** | **-2.45** | **7.15** | **7.56** | **10804** |



**Table 4: As in Table 2 but for differences between LEO IR-window AMVs and Aeolus Rayleigh-clear (RAY) winds in the Arctic (north of 60° N) and Antarctic (south of 60° S) polar regions.**

| Geographic Region | LEO Satellite | r | MCD (m s⁻¹) | SDCD (m s⁻¹) | RMSD (m s⁻¹) | Count |
|---|---|---|---|---|---|---|
| Arctic | Aqua | **0.89** | **-0.90** | **6.54** | **6.60** | **546** |
| Arctic | MetOp-A | 0.87 | -0.07 | 6.58 | 6.58 | 1530 |
| Arctic | MetOp-B | 0.87 | -0.11 | 6.61 | 6.60 | 1647 |
| Arctic | NOAA-15 | 0.88 | -0.12 | 6.78 | 6.77 | 249 |
| Arctic | NOAA-18 | 0.88 | -0.09 | 6.39 | 6.37 | 190 |
| Arctic | NOAA-19 | **0.84** | **-0.68** | **6.40** | **6.42** | **249** |
| Arctic | NOAA-20 | **0.91** | **-0.29** | **6.46** | **6.46** | **2642** |
| Arctic | S-NPP | **0.91** | **-0.32** | **6.49** | **6.49** | **1894** |
| Arctic | Terra | **0.88** | **-0.37** | **6.45** | **6.46** | **988** |
| Antarctic | Aqua | 0.92 | 0.28 | 8.22 | 8.22 | 1201 |
| Antarctic | MetOp-A | **0.88** | **-0.46** | **8.51** | **8.52** | **1471** |
| Antarctic | MetOp-B | **0.88** | **-0.59** | **8.36** | **8.37** | **1788** |
| Antarctic | NOAA-15 | 0.88 | -0.77 | 9.79 | 9.80 | 249 |
| Antarctic | NOAA-18 | 0.87 | -0.66 | 10.57 | 10.56 | 172 |
| Antarctic | NOAA-19 | 0.82 | -0.35 | 11.16 | 11.16 | 836 |
| Antarctic | NOAA-20 | **0.94** | **-0.73** | **8.17** | **8.20** | **2667** |
| Antarctic | S-NPP | **0.95** | **-0.76** | **7.49** | **7.53** | **2384** |
| Antarctic | Terra | 0.88 | -0.10 | 9.32 | 9.31 | 996 |




**Table 5: As in Table 4 but for Aeolus Mie-cloudy (MIE) winds.**

| Geographic Region | LEO Satellite | r | MCD (m s⁻¹) | SDCD (m s⁻¹) | RMSD (m s⁻¹) | Count |
|---|---|---|---|---|---|---|
| Arctic | Aqua | **0.94** | **-0.54** | **5.24** | **5.26** | **848** |
| Arctic | MetOp-A | **0.92** | **-0.54** | **5.61** | **5.63** | **3311** |
| Arctic | MetOp-B | **0.93** | **-0.56** | **5.09** | **5.12** | **3650** |
| Arctic | NOAA-15 | 0.93 | -0.30 | 4.92 | 4.92 | 375 |
| Arctic | NOAA-18 | **0.93** | **-0.72** | **5.08** | **5.13** | **336** |
| Arctic | NOAA-19 | 0.90 | -0.12 | 5.15 | 5.15 | 395 |
| Arctic | NOAA-20 | 0.94 | 0.03 | 5.17 | 5.17 | 5225 |
| Arctic | S-NPP | 0.94 | 0.09 | 5.30 | 5.30 | 4006 |
| Arctic | Terra | **0.94** | **-0.57** | **4.76** | **4.79** | **1689** |
| Antarctic | Aqua | 0.94 | 0.21 | 6.89 | 6.89 | 1065 |
| Antarctic | MetOp-A | **0.93** | **-0.44** | **7.51** | **7.52** | **1761** |
| Antarctic | MetOp-B | **0.93** | **-0.67** | **6.74** | **6.77** | **2142** |
| Antarctic | NOAA-15 | **0.95** | **-0.83** | **5.95** | **6.00** | **291** |
| Antarctic | NOAA-18 | 0.95 | -0.14 | 6.37 | 6.36 | 231 |
| Antarctic | NOAA-19 | 0.92 | -0.13 | 7.88 | 7.88 | 930 |
| Antarctic | NOAA-20 | **0.97** | **-0.72** | **6.31** | **6.35** | **4537** |
| Antarctic | S-NPP | **0.97** | **-0.73** | **6.40** | **6.44** | **4356** |
| Antarctic | Terra | 0.92 | -0.08 | 7.13 | 7.13 | 958 |

Differences in the SH extratropics and Antarctic pole exhibit higher SDCD values compared with the rest of the globe. This is

likely due to several factors. During the study period, the SH region of GEO fields-of-view covers a portion of the winter storm tracks that propagate eastward all the way around the Southern Ocean. The SH storm tracks exist year-round, and in winter (June-July-August) the upper-tropospheric subtropical jet is stronger and acts as a waveguide for eastward propagating baroclinic waves over a broader latitude range (Trenberth, 1991; Nakamura and Shimpo, 2004; Hoskins and Hodges, 2005), thus enhancing wind shear and storm track intensity. This is one factor that explains the higher SDCD values observed in GEO



differences in the SH extratropics, as AMV uncertainties tend to increase with increasing wind speed (Posselt et al., 2019) and high wind shear (Bormann et al., 2002; Cordoba et al., 2017). In the Antarctic polar region, the general strengthening of the polar vortex aloft in late winter/early spring (i.e., during the study period) is related to a stronger equator-pole temperature gradient brought about by gradually increasing subtropical lower stratospheric temperatures from March to September (Zuev and Savelieva, 2019). A stronger Antarctic polar vortex is associated with stronger zonal winds aloft (and thus stronger wind

shear) which would limit accurate AMV and Aeolus wind retrievals, thereby increasing the corresponding SDCD values on both accounts. Surface effects may also play a role, as very cold brightness temperatures at or near the polar surface may be misinterpreted as very high cloud tops due to the low temperature contrast between clouds and the surface snow or ice (Key et al., 2016).

Section 4.1 compares GOES-16 AMVs with Aeolus HLOS winds. We chose to examine GOES-16 AMVs because compared

with other GEO satellites, GOES-16 exhibits high correlations with Aeolus RAY (> 0.90) and MIE winds (> 0.94), relatively small SDCD (5.82-6.51 m s$^{-1}$ for RAY, 4.72-5.05 m s$^{-1}$ for MIE), and have the largest sample size from which to compute robust statistics (see Table 2 and Table 3). The other GEO satellites are not further examined as they exhibit larger SDCD (Meteosat-8 and -11), have a much smaller sample size (Himawari-8, INSAT 3D, Meteosat-8 and -11), or have transitioned out of main operations (GOES-15). Section 4.2 discusses the comparison of all available LEO AMVs with Aeolus HLOS

winds during the study period. This is done because compared to GEO, LEO AMVs from each satellite comprise a relatively small sample of collocated winds, and this would render any associated performance metrics unreliable. Further, unlike the suite of available GEO satellites where each observe a different region of the globe (except for small areas where the footprints of neighboring satellites overlap), each LEO satellite observes AMVs in the same polar regions and thus detects the same atmospheric motions.

**4.1 GOES-16 AMVs vs. Aeolus**

**4.1.1 Rayleigh-clear (RAY) comparisons**

Figure 4 presents density scatterplots that summarize the relationship of GOES-16 AMVs to RAY winds stratified by geographic region and AMV method to highlight the regional differences in IR (Fig. 4a, 4d, 4g), WVcloud (Fig. 4b, 4e, 4h), and WVclear AMVs (Fig. 4c, 4f, 4i). Sample statistics are based on Aeolus as the reference dataset and are displayed in the

lower right of each panel. Note that in the NH and SH extratropics, most collocations are found in the upper-right quadrant. HLOS winds in this quadrant are of positive sign, indicating westerly flow which is the dominant direction of motion in the general circulation in the extratropics. In the tropics, maximum densities are found in the lower-left quadrant (small negative HLOS) as well as the upper-right quadrant. The negative HLOS winds in the lower-left quadrant indicate easterly flow that represents the tropical trade winds at lower levels, while the positive HLOS winds in the upper-right quadrant represent more

westerly tropical flow at upper levels (e.g., see Fig. 5d).





AMVs correspond well with RAY winds with correlations of 0.88-0.9 in the extratropics and 0.94 in the tropics. Note that most of the collocations for each AMV channel type fall close to the one-to-one line that indicates a perfect match. The best match is for WVclear AMVs and RAY winds. This is expected, as it is a comparison of winds obtained by tracking upper-tropospheric features in clear air with Aeolus winds retrieved in nearby clear scenes, and clear scenes are more homogeneous

over time and space scales, which in turn implies smaller collocation differences. WVclear AMVs exhibit the smallest SDCD values in each geographic region, and these fall within the range of known speed SD and RMS of all GEO AMVs relative to radiosonde winds (see Table 1). The smallest WVclear SDCD are found in the tropics where mean HLOSV is smallest around 7 m s$^{-1}$. IR and WVcloud comparisons show similar relationships with larger MCD and SDCD estimates. The largest SDCD in WVclear AMVs are found in the SH extratropics where mean HLOSV is fastest around 20 m s$^{-1}$. Higher wind speeds (and

most probably stronger wind shear) observed in the SH extratropics for all AMV channels can reduce the accuracy of both AMV derivations and Aeolus wind retrievals. The results suggest that the relationship of AMVs to RAY profiles characterizes known AMV uncertainty in clear scenes in the tropics and NH extratropics similar to high quality sources of wind profile observations. The certainty of such a statement for motions in the SH extratropics is more difficult to verify, as there are much fewer radiosonde observations available in the Southern Hemisphere for comparison (e.g., Durre et al., 2006, their Fig. 1).

Further, it can be inferred that QC'd WVclear AMVs represent well the dynamical flow in clear scenes in the tropics, particularly in summer where there is high moisture content available for tracking upper-level clear-sky water vapor features (Velden et al., 1997).

We next examine the vertical variation of AMV minus RAY winds (Fig. 5). This perspective has the potential to provide additional insight into how well each AMV channel represents the horizontal flow at various vertical levels. Mean vertical

profiles of QC'd GOES-16 AMV and RAY HLOSV and their mean differences and SDCD are plotted per AMV channel type in the NH extratropics (Fig. 5a-c), tropics (Fig. 5d-f), and SH extratropics (Fig. 5g-i). In Fig. 5a, 5d, and 5g, profiles of mean HLOSV for AMVs (solid lines) and Aeolus (dashed lines) show good agreement. In Fig. 5b, 5e, and 5h, mean differences in HLOSV (solid lines) and standard deviations of the differences (long dashed lines) are shown as well as the mean Aeolus L2B uncertainty (short dashed lines), with open circles indicating pressure levels at which HLOSV differences are statistically

significant at the 95% level (p-value < 0.05) using the paired two-sided Student's $t$-test. Corresponding collocation counts are shown in Fig. 5c, 5f, and 5i.

Vertical profiles of AMV and RAY HLOSV closely match in each geographic region. HLOSV increases with height, with NH speeds peaking at 20 m s$^{-1}$ near the jet stream level (~250 hPa) and decreasing aloft, and with SH speeds continuously increasing with height to more than 30 m s$^{-1}$ at the highest levels. Large HLOSV in the extratropics at upper levels correspond to a deep

layer of enhanced vertical wind shear associated with the corresponding jets and storm tracks captured in the northern and southern regions as viewed by GOES-16; the larger HLOSV in the SH indicate faster motions and stronger shear in the winter hemisphere.

**Figure 4: Density scatterplot of collocated GOES-16 AMVs and RAY winds. Rows are for the (a-c) NH extratropics (30-60° N), (d-f) tropics (TR) (30° S to 30° N), and (g-i) SH extratropics (30-60° S). Columns are for different AMV channel types: (left) IR, (center) WVcloud, and (right) WVclear. Colors indicate total number of collocated observation pairs within the cells plotted, which are 1 ms⁻¹ on a side. Sample statistics are displayed in the bottom right corner of each panel. Horizontal and vertical zero lines are plotted in black, as is the diagonal one-to-one line. Red star denotes statistical significance at 95% using the paired two-tailed Student's t-test. HLOSV units are m s⁻¹.**






**Figure 5: Vertical comparisons of collocated GOES-16 AMVs and RAY winds. The top row shows the NH extratropics (30-60° N), (a) mean AMV (solid lines) and RAY (dashed lines) winds (m s⁻¹), (b) MCD (solid), SDCD (long dashed), and Aeolus L2B uncertainty (short dashed) (m s⁻¹), and (c) collocation counts. (d-f) as in (a-c) but for the tropics (30° S to 30° N), and (g-i) as in (a-c) but for the SH extratropics (30-60° S). Colors denote AMV channel type: IR (red), WV cloud (blue), and WV clear (green). Colored open circles indicate levels where MCD are statistically significant at the 95% level (p-value < 0.05) using the paired Student's t-test. Vertical zero lines are displayed in the left and center panels in black. Levels with observation counts > 25 are plotted.**

HLOSV differences are statistically significant throughout the vertical in all geographic regions and exhibit similar vertical behavior in both extratropical regions and opposing behavior in the tropics. In the NH and SH extratropics at levels where collocation counts peak, mean HLOSV collocation differences are small (-1.0 m s⁻¹ to -2.0 m s⁻¹) yet statistically significant





and could represent a small slow AMV bias which has been previously noted by Bormann et al. (2002). SDCD are constant in the vertical around 6 m s$^{-1}$ in the NH; SDCD in the SH are larger (6-8 m s$^{-1}$) and increase with height. Aeolus L2B uncertainty represents about half of the SDCD and is smallest for WVclear winds (2.5-3 m s$^{-1}$) and largest for IR and WVcloud winds (3-4 m s$^{-1}$) in all geographic regions. In the tropics, winds speeds peak in intensity around 300-400 hPa, suggesting enhanced wind shear within that layer. A small fast bias, that is the positive MCD of 0.5-1.0 m s$^{-1}$, could be due to height assignment

errors in layers of high winds and enhanced vertical wind shear (Cotton et al. 2020). IR and WVcloud collocation counts in the tropics peak around 200 hPa and depict higher cloud-top motions related to the ITCZ, e.g., possibly thin anvil cirrus, which can hinder accurate height assignments to the AMVs. Moreover, the Aeolus QC acts to retain winds with larger uncertainties at levels above 200 hPa. This would explain the corresponding increase in SDCD at these levels. In all regions, WVclear AMVs exhibit smaller SDCD relative to the other channel types.

The findings confirm that QC'd GOES-16 AMVs exhibit relatively small MCD (with magnitudes less than 1.0 m s$^{-1}$) with respect to QC'd Aeolus RAY winds, with SDCD values (5-7 m s$^{-1}$) that are close to AMV error values relative to radiosondes/rawinsondes (see Table 1). AMV MCD and SDCD relative to Aeolus winds are found to be dependent on AMV method, geographic region, and vertical layer, in agreement with the findings in Velden et al. (1997) and Posselt et al., (2019). SDCD values tend to be larger in the SH extratropics where upper-level winds associated with the stronger subtropical jet

storm track in winter are enhanced (Trenberth, 1991; Hoskins and Hodges, 2005). Among the three AMV methods examined, WVclear AMVs perform best with respect to Aeolus RAY winds and exhibit small yet significant HLOSV differences that generally match AMV performance metrics with respect to high quality radiosonde winds. Although we compare mean AMV-Aeolus collocation differences with speed statistics, it should be noted that in general, the HLOS wind generally approximates the zonal component of the horizontal flow (away from the poles) rather than the wind speed. However, the magnitude of the

mean HLOS wind can act as a good approximation of the wind speed in regions where the mean flow is more zonal (e.g., away from the poles).

### 4.1.2 Mie-cloudy (MIE) comparisons

AMVs more closely match collocated MIE winds compared to RAY, due in part to smaller MIE random errors. Figure 6 presents density scatterplots similar to those in Fig. 4 but compares AMV and MIE winds. Only IR and WVcloud AMV

collocations are shown, as MIE winds represent cloudy scenes and WVclear channels are tuned to track moisture features in clear skies.

IR and WVcloud winds correspond very well with MIE winds; the collocations exhibit high correlations ranging from 0.93 in the NH extratropics to 0.97 in the tropics. Most MIE collocations fall along the one-to-one line that corresponds to a perfect match. MCD and SDCD are smallest in the tropics at -0.3 m s$^{-1}$ and 4.4-4.6 m s$^{-1}$, respectively, particularly for WVcloud

comparisons which seem to have the fewest outliers. SH extratropical comparisons exhibit the largest SDCD (5-5.2 m s$^{-1}$) which still fall within range of those associated with high quality radiosonde winds (Velden and Bedka, 2009; Santek et al., 2019). Corresponding Aeolus L2B uncertainty represents almost 50% of the MIE SDCD and is generally constant (2 m s$^{-1}$),





particularly at levels above 400 hPa, in all geographic regions. MIE comparison SDCD and L2B uncertainty are considerably smaller than those for RAY comparisons, and this could be attributed to a combination of better correlated AMVs derived

from motions in cloudy scenes and the higher accuracy of Aeolus MIE wind retrievals. The mean collocation distances for MIE winds (~51 km) is only slightly smaller than for RAY winds (~60 km) and should not contribute much to the smaller MIE SDCD.

The smaller SDCD observed in the NH and tropics suggest that AMVs represent well the cloud-tracked motions associated with the North Atlantic storm track in summer and the summer-shifted ITCZ; such features are well-defined by high MIE

number densities in the north and middle portions of the GOES-16 field-of-view in Fig. 1b. The larger SH SDCD suggest reduced accuracy in AMV winds that could be due to height assignment errors in regions of higher wind speed and shear associated with stronger SH winter storm tracks.

Figure 7 depicts the vertical distributions of HLOSV and the differences between AMV and MIE winds. Compared to the RAY collocations, the MIE collocations show virtually identical profiles of HLOSV speeds for the IR and WVcloud samples

but different vertical distributions of the differences as well as smaller SDCD in each geographic region. Significant negative MCD are largest at -1.5 m s$^{-1}$ at mid-levels in the tropics and -2.0 m s$^{-1}$ at upper-levels in the NH extratropics, respectively. However, some of the larger differences occur at levels with a small sample size and may not be reliable. In the tropics, the largest MCD correspond to winds below the higher cloud-tops of the ITCZ and could be attributed to height assignment errors combined with high winds and enhanced vertical wind shear in the region. Despite the vertical variation of the MCD, SDCD

are relatively constant at 4-5 m s$^{-1}$ in the tropics. In the NH extratropics, SDCD appear to slightly increase with height corresponding to AMV and Aeolus winds in regions of high wind speeds and enhanced wind shear.

In the SH extratropics, statistically significant HLOSV differences are relatively small at low levels (-1.0 m s$^{-1}$) and dramatically become more negative with height, reaching values that are more negative than -3.0 m s$^{-1}$ above 300 hPa. It can be inferred that the high wind velocities at upper levels (Fig. 7g) exemplify an intensified jet and corresponding enhanced

vertical wind shear in the region of the SH winter storm tracks. The large differences aloft exhibit relatively small increases in SDCD with height, which are on the order of 5-6 m s$^{-1}$ throughout the vertical. The larger mean differences represent over 8.5% of the corresponding high wind speeds at upper levels and could be attributed to larger height assignment errors in the region of corresponding enhanced vertical wind shear.

Statistics of AMV minus MIE collocation differences are consistent with those for AMV comparisons with high quality

radiosonde winds. Good quality GOES-16 AMVs exhibit significant small MCD and SDCD in cloudy scenes. IR and WVcloud AMVs perform best in the tropics, and exhibit similar MCD and SDCD throughout the vertical in the tropics and NH extratropics. SDCD for MIE comparisons are considerably smaller than those for RAY winds, suggesting that RAY errors are greater than MIE errors, and in turn contribute more to the RAY SDCD than for MIE comparisons. MIE MCD exhibit unique behavior in the SH that highlights the known slow bias in the extratropical upper atmosphere. MCD in the SH extratropics are

significant and large relative to the other geographic regions, and corresponding SDCD are larger and increase with height.



The larger MCD and SDCD above 300 hPa suggests that cloud-tracked upper-level motions are more difficult to quantify if they exist near levels of enhanced baroclinic instability, e.g., the extratropical SH winter storm tracks at jet stream level.

**Figure 6: As in Figure 4 but for comparisons of IR (left) and WVcloud (right) AMVs and MIE winds.**



*Figure 7: As in Figure 5 but for comparisons of IR (red) and WV cloud (blue) AMVs and MIE winds.*

**4.2 LEO AMVs vs. Aeolus**

Figure 8 presents density scatterplots that compare LEO AMVs derived from IR window channels with RAY and MIE winds in the Arctic (Fig. 8a-b) and Antarctic polar regions (Fig. 8c-d). LEO AMVs correspond well with both Aeolus observing modes in the polar regions. Comparisons in the Arctic have small yet significant mean differences in HLOSV (around -0.2 m $s^{-1}$) and SDCD estimates of 5.2-6.5 m $s^{-1}$, while Antarctic comparisons exhibit larger MCD and SDCD. Further, Arctic MIE





comparisons exhibit the smallest SDCD, and Antarctic RAY comparisons have the largest SDCD and more evident outliers.

The results suggest that during the period of study in the Arctic, IR LEO AMVs are best able to capture cold scene cloud-tracked motions during the summer season when cloudiness increases in the vertical and more water vapor content is generally available to track features (Alekseev et al., 2018). Water vapor content in the Arctic is largest in summer due to an influx of water vapor from melting ice and snow and receding sea ice extent as well as enhanced meridional moisture fluxes from low latitudes (Alexseev et al., 2018).

As for the GOES-16 case study, we examine the mean differences in the vertical between all LEO AMV winds and Aeolus winds to ascertain how well AMVs characterize the dynamical flow at the poles (Fig. 9). RAY (red colors) and MIE (blue colors) comparisons are presented together. Figures 9a and 9d display the mean AMV and Aeolus HLOSV in the Arctic and Antarctic, respectively, and Fig. 9b and 9e show the MCD and SDCD. Corresponding observation counts are shown in Fig. 9c and 9f.

AMV and Aeolus winds exhibit similar speeds that increase with height in both polar regions. In the Arctic (boreal summer), MIE winds and collocated AMVs show faster motions relative to RAY comparisons in mid- to upper-levels. Statistically significant MCD are on the order of -0.5 m s$^{-1}$ at mid-levels where collocation counts peak, representing slower AMV winds relative to Aeolus. The mean differences become larger (more negative) nearer the tropopause (~300-250 hPa) where speeds reach upwards of 15 m s$^{-1}$. SDCD in the Arctic are relatively constant throughout the troposphere with smaller MIE SDCD

(~5 m s$^{-1}$) and Aeolus L2B uncertainty (1-2 m s$^{-1}$) relative to RAY (~7 m s$^{-1}$ and 3-4 m s$^{-1}$, respectively), suggesting the higher accuracy of MIE winds. The near doubling of MIE collocation counts at mid-levels relative to RAY could be due to increased cloudiness associated with more moisture availability in Arctic summer (Alekseev et al., 2018).

In the Antarctic (austral winter), wind speeds increase from 5 m s$^{-1}$ at mid-levels to nearly 30 m s$^{-1}$ at very high levels (~150 hPa), and RAY comparisons are shown to capture generally faster motions throughout the vertical column. MCD are small

(around -0.5 m s$^{-1}$) at levels where collocation counts peak and become larger aloft in the layer of sharply increasing winds where they represent over 10% of the corresponding wind speeds. As in the Arctic, MIE comparisons in the Antarctic have smaller SDCD (6-7 m s$^{-1}$) and L2B uncertainty (2 m s$^{-1}$) than RAY (8-12 m s$^{-1}$ and 4-5 m s$^{-1}$, respectively) throughout the vertical, but both exhibit larger uncertainties in the Antarctic. Further, RAY SDCD increase with height from ~7 m s$^{-1}$ at low levels to over 10 m s$^{-1}$ at very high altitudes; similarly, corresponding Aeolus RAY L2B uncertainties increase from 4-5 m s$^{-1}$

$^{1}$. Larger MCD aloft in the Antarctic could be attributed to the lower accuracy of AMV height assignments in the layer of increasing wind speed and corresponding enhanced vertical wind shear related to the strengthening of the Antarctic polar vortex in late winter/early spring. Higher SDCD values at upper levels may be attributed to the inclusion of Aeolus winds with larger uncertainties above 200 hPa following the QC as well surface effects, as very cold brightness temperatures near the surface may be misinterpreted as high clouds (Key et al., 2016).

The relationship of LEO IR AMVs to Aeolus winds depends on the polar region and Aeolus observing mode, Rayleigh or Mie. Overall, polar AMVs have smaller MCD relative to MIE than RAY. In Arctic summer, more water vapor is available to track features throughout the vertical column, and this results in similar MCD and SDCD with respect to both Aeolus observing



modes. Differences are small and significant for MIE comparisons, and SDCD are generally smaller than for RAY comparisons by 1-2 m s$^{-1}$. In Antarctic winter, MCD and SDCD become larger with height. This behavior may be partially due to the

mischaracterization of cold surfaces as clouds and partially due to strong wind shear aloft related to a strengthening of the Antarctic polar vortex, which would diminish the representativeness of AMVs and Aeolus winds as well as reduce the accuracy of corresponding height assignments.

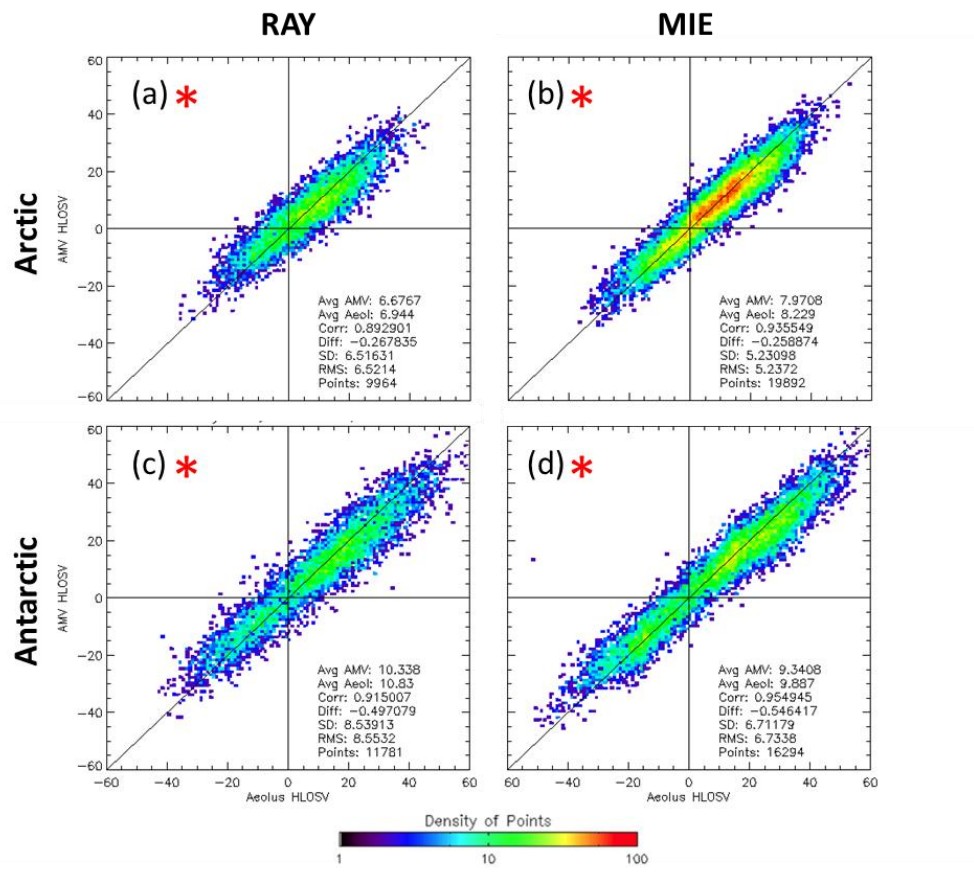

**Figure 8: Density scatterplot of collocated IR-window AMVs from all available LEO satellites and RAY (left column) and MIE (right column) winds: Comparisons in (a-b) Arctic (north of 60° N), and (c-d) Antarctic (south of 60° S). Colors indicate total number of collocated observation pairs within the cells plotted, which are 1 ms$^{-1}$ on a side. Sample statistics are displayed in the bottom right corner of each panel. Horizontal and vertical zero lines are plotted in black, as is the diagonal one-to-one line. Red star denotes statistical significance at the 95% level using the paired two-tailed Student's t-test. HLOSV units are m s$^{-1}$.**

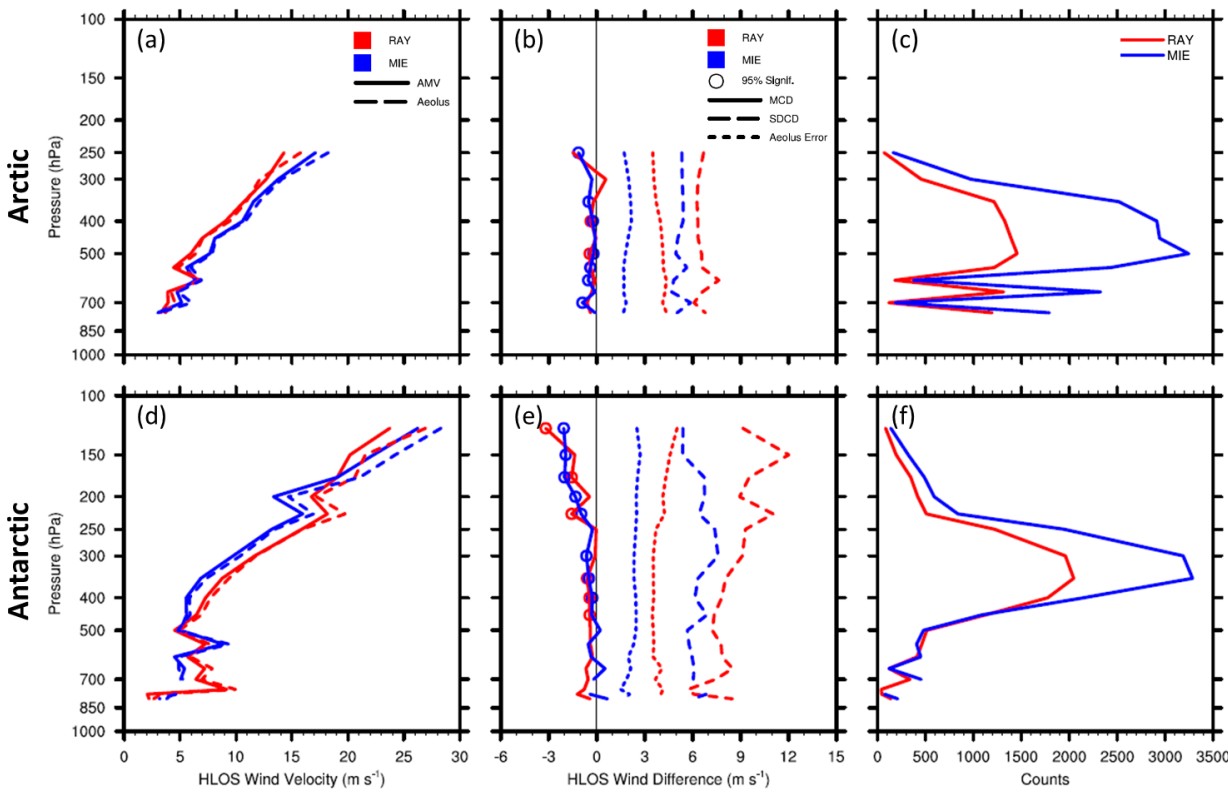


**Figure 9: Vertical comparisons of collocated LEO AMVs and RAY (red) and MIE (blue) winds. The top row shows the Arctic (north of 60° N), (a) mean AMV (solid lines) and Aeolus (dashed lines) winds (m s⁻¹), (b) MCD (solid), SDCD (long dashed), and Aeolus L2B uncertainty (short dashed) (m s⁻¹), and (c) collocation counts. (d-f) as in (a-c) but for the Antarctic (south of 60° S). Colored open circles indicate levels where MCD are statistically significant at the 95% level (p-value < 0.05) using the paired Student's t-test.**
**Vertical zero lines are displayed in the center panels in black. Levels with observation counts > 25 are plotted.**

## 5 Summary and conclusions

This study summarizes statistical comparisons of AMVs with the novel Aeolus L2B HLOS winds based on specific sets of conditions and discusses their relationship to known AMV characteristics. Because Aeolus observes the HLOSV—the horizontal wind projected onto the HLOS of the DWL—derived from the detection of molecular and aerosol backscattering

signals, the assessments of mean collocation differences (AMV minus Aeolus) and SD of the differences (MCD and SDCD) are all in terms of AMV winds projected onto the collocated Aeolus HLOS. In the tropics, due to the Aeolus observing geometry, HLOSV represents the zonal wind. Aeolus HLOSV profiles utilized in this study are classified as RAY or Rayleigh-clear winds (representing mostly clear-sky scenes) and MIE or Mie-cloudy winds (representing cloudy scenes only). Quality controls recommended by ESA for Aeolus and by the user community for the satellite winds are applied. Quality controlled

winds from each dataset are retained for analysis.





The performance of quality controlled AMVs relative to collocated Aeolus winds of quality controlled are characterized by analyzing the sample statistics of the collocated differences, AMV minus Aeolus HLOSV. These statistics should not be strictly interpreted as overall AMV performance, as differences arise from errors in both AMVs and Aeolus winds and from representativeness and collocation errors. Comparisons of GOES-16 AMVs and IR window cloud-track AMVs from LEO
satellites are assessed to estimate the dependence of AMVs on different combinations of conditions including Aeolus observing mode/scene type (clear or cloudy), AMV derivation method (IR window, WVcloud, and WVclear), and geographic region (tropics and extratropics for GOES-16, Arctic and Antarctic polar regions for LEO). Vertical distributions of differences in HLOSV are examined, as this perspective has the potential to provide additional insight into how well each band for AMV retrievals represents the horizontal flow in the vertical.

Relative to Aeolus, AMV performance metrics exhibit different characteristics in clear and cloudy scenes that vary with geographic region and in the vertical, in agreement with the findings in Velden et al. (1997) and Posselt et al. (2019). Overall, GEO and LEO AMVs are found to correspond very well with Aeolus RAY and MIE winds, and on average GEO AMV minus Aeolus MCD and SDCD values fall within the ranges of known biases and uncertainties of AMVs. MCD are small, and SDCD and mean collocation distances are smaller for MIE compared to RAY collocations, reflecting the higher accuracy of MIE
winds and of AMVs in cloudy scenes and possibly larger collocation errors for the RAY winds which tend to have larger collocation distances with AMVs relative to MIE. Larger SDCD are evident in the SH where wind shear is enhanced especially in winter due to the strengthening of the Antarctic polar vortex (Zuev and Savelieva, 2019), which can affect the representativeness of both AMVs and Aeolus winds. In the Arctic, MIE comparisons exhibit small MCD and SDCD consistent with known LEO AMV characteristics, while Antarctic RAY and MIE comparisons show generally larger SDCD.

GOES-16 and LEO AMV MCD and SDCD characterizations based on Aeolus winds are summarized. GOES-16 was chosen as a representative of GEO performance, as the AMVs exhibit high correlations with Aeolus, relatively low MCD and SDCD, and have a large sample size from which to compute robust statistics. The summary assessment of all LEO AMVs provides a unique, comprehensive perspective on the characteristics of polar AMVs using a larger sample of collocated Aeolus wind profiles relative to other available datasets, e.g., radiosonde profile data.

GOES-16 AMVs are found to correspond well with RAY and MIE winds. In clear scenes in the tropics and NH extratropics, MCD are small and SDCD fall within the accepted range of AMV error values when compared with high quality radiosonde winds (Velden and Bedka, 2009; Santek et al., 2019). WVclear AMVs perform best with respect to Aeolus RAY winds, with smaller MCD (-0.6 m s$^{-1}$ to 0.3 m s$^{-1}$) and SDCD values (5.4-5.6 m s$^{-1}$) relative to other AMV channel types. Relative to MIE winds, IR window cloud-track and WVcloud AMVs exhibit similar MCD in the tropics (-0.3 m s$^{-1}$) and NH extratropics (-0.5
m s$^{-1}$ to -0.7 m s$^{-1}$), with WVcloud AMVs exhibiting the smallest SDCD. MIE collocations have smaller SDCD (~4.4 m s$^{-1}$ to 4.8 m s$^{-1}$) and corresponding Aeolus L2B uncertainty (1-2 m s$^{-1}$) than RAY collocations. In the SH extratropics, MCD are large relative to the other geographic regions, and SDCD estimates are larger and increase with height, suggesting that cloud- and moisture-tracked motions related to extratropical SH winter storm tracks may be more difficult to accurately quantify in the SH, particularly near the jet stream level.



The relation of LEO IR AMVs to Aeolus winds differs between RAY and MIE comparisons (i.e., in clear vs cloudy scenes) and varies with polar region and Aeolus observing mode. Overall, MIE comparisons have smaller MCD compared to RAY. In Arctic summer, MIE MCD are small but statistically significant, and SDCD and corresponding Aeolus L2B uncertainties are generally smaller than for RAY collocations by 1-2 m s$^{-1}$. In Antarctic winter, mean AMV and Aeolus winds sharply increase with height. The corresponding MCD become more negative and SDCD are larger than in the Arctic and also increase

with height. The larger MCD aloft could be attributed to the lower accuracy of AMV height assignments in regions of high winds and strong wind shear related to the strengthening of the Antarctic polar vortex in late winter/early spring, as well as the possible mischaracterization of very cold surface temperatures as clouds. The inclusion of Aeolus winds with larger uncertainties above 200 hPa may also play a role.

The findings presented here provide information on the variation of AMV characteristics relative to Aeolus RAY and MIE
winds, and suggest that Aeolus could be used as a standard for the comparative assessment of AMVs pending additional bias corrections to the Aeolus L2B winds. Comparisons with Aeolus HLOS winds provide estimates consistent with known AMV bias and uncertainty in the tropics, NH extratropics, and in the Arctic, and at mid- to upper-levels in both clear and cloudy scenes. WVclear winds perform best relative to RAY winds. Comparisons between IR and WVcloud AMVs and Aeolus MIE winds reveal similar MCD and significant smaller SDCD in the NH and tropics with respect to RAY comparisons. SH
comparisons generally exhibit larger SDCD that could be attributed to height assignment errors in regions of high winds and enhanced vertical wind shear. The level of agreement between AMVs and Aeolus winds varies per stratification including the Aeolus observing mode coupled with AMV derivation method, geographic region, and height of the collocated winds. These combinations of conditions should be considered in future comparison studies and impact assessments involving 3D winds. Additional corrections to the Aeolus dataset, e.g., via the removal of DWL instrument calibration-dependent error or a bias
correction utilizing Total Least Squares regression as discussed in Liu, H. et al. (2021), are anticipated to further refine the results.

The use of Aeolus HLOS winds as a benchmark dataset has valuable implications for future endeavors involving validation of 3D winds and the use of such data in NWP. For example, these findings contribute to the ongoing development of a feature track correction (FTC) observation operator to account for AMV height assignment and other biases in data assimilation
(Hoffman et al., 2021). One lesson learned from this study is that QC of both AMV and Aeolus observations is critical and largely improves the results.

**Data availability**

The Aeolus L2B Earth Explorer data used in this study are publicly available and can be accessed via the ESA Aeolus Online Dissemination System (https://aeolus-ds.eo.esa.int/oads/access/). The NCEP SATWND BUFR AMV dataset can be provided
by the corresponding author (katherine.lukens@noaa.gov) upon request.





**Author contributions**

Kevin Garrett and Kayo Ide proposed the project as co-investigators and provided expertise that guided this work. Brett Hoover and David Santek developed the collocation algorithm used. Katherine E. Lukens performed most of the work that included the implementation of the collocation algorithm and comparison analysis. David Santek and Ross N. Hoffman provided
additional intellectual support that considerably improved the article. Katherine E. Lukens prepared the manuscript with contributions from all co-authors.

**Competing interests**

The authors declare that they have no conflict of interest.

**Acknowledgements**

The authors would like to thank their colleagues at NOAA/NESDIS/STAR for overseeing this work as well as Dr. Mike Hardesty of CSU/CIRA, Dr. Iliana Genkova of IM Systems Group, Inc. (IMSG), and the rest of the US Aeolus Cal/Val team for guidance and support. The University of Wisconsin-Madison S4 supercomputing system (Boukabara et al., 2016) was used in this work. The authors acknowledge support from the NOAA/NESDIS Office of Projects, Planning, and Acquisition (OPPA) Technology Maturation Program (TMP) through CICS and CISESS at the University of Maryland/ESSIC [NA14NES4320003
and NA19NES4320002] and CIMSS at the University of Wisconsin-Madison [NA20NES4320003]. The scientific results and conclusions, as well as any views or opinions expressed herein, are those of the authors and do not necessarily reflect those of NOAA or the U.S. Department of Commerce.

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
