# Peer review of "Exploiting Aeolus Level-2B Winds to Better Characterize Atmospheric Motion Vector Bias and Uncertainty"

_Atmospheric Measurement Techniques, 2021_

## Author Comment (AC3)

**Response to comments of Anonymous Referee #1 on *AMT* preprint "Exploiting Aeolus Level-2B Winds to Better Characterize Atmospheric Motion Vector Bias and Uncertainty" by Katherine E. Lukens et al. Atmos. Meas. Tech. Discuss., https://doi.org/10.5194/amt-2021-277-RC1, https://doi.org/10.5194/amt-2021-277-RC2, https://doi.org/10.5194/amt-2021-277-RC3, 2021**

Thank you very much for your careful and helpful comments. We have revised our manuscript following your suggestions. We greatly appreciated the opportunity to iterate with you on some of your comments during the discussion period.

[Below, quotes from your comments are repeated verbatim in *green italic*. All your comments from the discussion are included here. Comments made by us and by you that are resolved are included here but are indented. Changes in quoted text are indicated as additions and . Comments from author to author are in this color.]

*General comments*

*The paper is important and covers an important Task since Aeolus was launched: the comparison of Aeolus Winds with AMVs, for the continuous improvement of both sources of wind observations. The paper should be accepted after some a minor revision/correction.*

Thank you.

*Please consider all following items.*

We have.

*Specific comments - Important comments*

*1a. Please include a description, including a formula, of the Statistics used in your paper (MCD, SD, SDCD,…; possibly also the Speed Bias, Speed SD, RMS, Vector Diff, Vector RMS included in Table 1), and their relationship with the Statistics used as Standard procedure for AMV validation defined by the International Winds Working Group in its 1996 Workshop (http://cimss.ssec.wisc.edu/iwwg/iww3/index_3rdWindsWorkshop.htm) in following report: http://cimss.ssec.wisc.edu/iwwg/iww3/p17-19_WGReport3.pdf.*

This is a good point. Following our discussion with you in the open forum, we now add,

"Because we are comparing the AMV and Aeolus HLOSV, a scalar quantity, our statistics can only be analogs of the standard one. We include the formulae for all the statistics in Appendix A."

And we have added that appendix.

*This is important; if I look throughout the internet "Mean collocation differences (MCD)", I find very few references, and all of them are from the year 2021.*

We define MCD when first used and its computing formula is included in Appendix A.

*1b. Why were the original Statistics for winds in this IWW3 Report not used?*

See our response to 1a.

*1c. The collocation criteria used between AMVs and Aeolus winds in lines 180-183 are also different to those defined by the International Winds Working Group in its 1998 Workshop (https://cimss.ssec.wisc.edu/iwwg/iww4/index_4thWindsWorkshop.htm) in following report: https://cimss.ssec.wisc.edu/iwwg/iww4/p19-20_WGReport3.pdf. Please comment on the differences on both criteria, and if you could expect some impact in using one or the other procedure.*

The manuscript now states, "Our choice of collocation criteria is conservative compared to those defined by the IWWG 1998 workshop (Velden and Holmlund 1998). Although the larger time and distance criteria defined by IWWG (90 vs 60 minutes and 150 vs 100 km) might retain more collocation pairs and thus a larger sample, the collocated winds would more likely have larger MCD and SDCD. Our smaller time and distance criteria restrict the number of possible Aeolus matches to any one AMV and help avoid Aeolus matches from two different orbits. The IWWG height criterion is a fixed pressure difference (25 hPa) that might be too small at lower levels where pressure layers are tightly spaced in elevation but too large in the upper atmosphere where the elevation distance between pressure layers is much larger. Our height criterion is based on a $\log_{10}$ scale and accounts for the varying distances between pressure layers throughout the vertical and corresponds to pressure differences ranging from approximately 300 to 1 hPa for pressures from 1000 to 10 hPa, respectively."

*2a. There is an important error in lines 48-50: with the text: "(1) water vapor cloud-top (WVcloud) channels are used to track upper-level cloud top motions, and (2) water vapor clear-sky (WVclear) channels are used to detect upper-tropospheric features (e.g., jet stream and waves) by tracking water vapor motions in clear air you seem to say that some WV channels are used for Clouds and other ones for Clear sky, and this is completely wrong. The same channel (f.ex. WV062) can be used to calculate Cloud AMVs (tracking clouds) and Clear air AMVs (tracking moisture patterns in other parts of the image). Please correct the text.*

Done. The manuscript now states,

"Infrared bands that are specifically sensitive to water vapor (WV) absorption can capture different atmospheric motions using the same channel by tracking in two ways: (1) water vapor cloud-top (WVcloud) channels are used to track upper-level cloud-top motions, and (2) water vapor motions in clear air water vapor clear-sky (WVclear) channels are used to detect related to upper-tropospheric features (e.g., including the jet stream and atmospheric waves) by tracking water vapor motions in clear air (Velden et al., 1997)."

*2b. Related to the previous comment, please change throughout all the text all expressions "AMV channel" and "AMV channel type" to "AMV type", to avoid the same error. And for example change in line 142-143: "IR, WVcloud, and WVclear channels" to "IR, WVcloud, and WVclear AMV types". Please be careful with this.*

The manuscript now refers to type everywhere.

*3. Lines 73-74: "Such a direct comparison has not previously been possible due to the sparse spatial coverage of other available reference datasets, e.g., rawinsonde winds". I am not so sure you cannot use rawinsonde winds for this; please provide more detail to this sentence, so that you can conclude what you say.*

Agree. The manuscript now states, "Such a direct global comparison has not previously been possible due to the  limited spatial coverage of other available reference datasets, e.g., rawinsonde winds, which are mostly available in the Northern Hemisphere over land."

*4a. Line 110: please try to give more detail or reference about why in spite of the "M1 bias correction", there are still some "remaining biases" which you mention in line 26.*

Done. The manuscript now states,

"While the M1 bias correction is capable of considerably reducing the telescope-induced wind bias, some residual bias may remain, e.g., in cases where the top-of-atmosphere reflected radiation strongly influences the telescope temperature (Weiler et al., 2021). Additionally, residual biases may remain in part due to potential calibration issues of the Aeolus L2B winds that could in turn lead to biases between Aeolus and NWP background winds (Liu et al., 2022)."

*4b. Line 110: please give some detail or reference about the "Quality Controls" used in Aeolus.*

Done. The manuscript now states, "They found that with the application of the M1 bias correction and proper quality controls (QC) (see Rennie and Isaksen (2020a) or Section 3 for specific QC criteria used in this paper) as well as Aeolus black-listed dates taken into account…".

*4c. Line 110: please give some detail or reference on why there are "Aeolus black-listed dates".*

Done. The manuscript now states, "… Aeolus blocklisted period (defined as a period of time when the Aeolus dataset is known to be degraded and should not be included in research or operations).".

*5a. What is the "horizontal/vertical accumulation lengths" you mention in lines 198-199?*

The manuscript now states, "Horizontal and vertical accumulation lengths refer to the horizontal and vertical distances over which individual measurement signals are accumulated and averaged to improve the signal-to-noise ratio. In this way, the Aeolus observations represent wind volumes and not discrete points or levels. The accumulation lengths can vary and depend on the processor settings."

*5b. What is the "L2B uncertainty" you mention in line 201?*

The manuscript now states, "L2B uncertainty refers to the Aeolus HLOS wind error estimate assigned to each wind measurement."

*6. Differences in time and location between the two observations can be not so important as the ones you mention in lines 230-231: "differences due to collocation (i.e., due to different times and locations 230 of the two observations) could play a role in increasing the differences between the collocated HLOS winds". Please check as an example: "Chapter 2.3 IMPACT OF THE REPRESENTATIVITY OF THE RADIOSOUNDING WINDS" in following reference: https://www.nwcsaf.org/AemetWebContents/ScientificDocumentation/Documentation/GEO/v201 6/NWC-CDOP2-GEO-AEMET-SCI-VR-Wind_v1.0.pdf. Then comparatively check how much the wind can change considering the time and distance implied in your collocations, and update your sentence if needed.*

We have deleted this sentence.

*7. Important: In chapter 4.1.2 line 419 you say you compare only IR cloud AMVs and WV cloud AMVs with Aeolus Mie Cloud winds. Why don't you do the same in chapter 4.1.1, comparing Aeolus Rayleigh Clear Air winds with WV clear air AMVs only? Please include in the text an explanation of why you are acting differently in both chapters. Evaluate also if there could be two different elements here, which behave differently:*

*- flow related to cloud features, evaluated by both Aeolus Mie winds and cloudy AMVs,*

*- and flow related to clear air features, evaluated by both Aeolus Rayleigh winds and clear air AMVs.*

We now add, "To increase the size of our collocation data set, we compared all types of GOES-16 AMVs to both Rayleigh-clear and Mie-cloudy winds. In addition, we do not show results from WVclear AMV collocations with Mie-cloudy winds as correlations for this category of collocations are poor and the sample size is very small (see Table 1), and this result may be unreliable. With a larger data set it might be possible to compare Rayleigh-clear and Mie-cloudy winds to clear and cloudy AMVs only, respectively."

In RC2 you wrote:

*This sentence is better, but I think it still misses some indication to the fact that wind observations obtained from clear air and cloud AMVs do not behave exactly in a similar way. For example, it is a recognized fact inside the AMV community that time scales of winds related to clear air AMVs are longer (30-60 minutes) than the time scales of winds related to cloud AMVs (10-15 minutes), and this is enough so that show some differences.*

*The discussion included the following exchange:*

> Thank you for making the point about AMV time scales. We will add additional information on AMV time scales to the manuscript, and we plan to cite one or more of the following supporting references. Can you recommend any peer reviewed references that could be used?
>
> De Smet, A., 2002: Operational AMV products derived with Meteosat-6 rapid-scan data. Proc. of the Sixth Int. Winds Workshop, Madison, WI, WMO, 179-185. Available online at http://cimss.ssec.wisc.edu/iwwg/iww6/session4/deSmet.pdf.
>
> Schmetz, J., K. Holmlund, H.P. Roesli, and V. Levizzani, 2000: On the Use of Rapid Scans, Proceedings of the Fifth International Winds Workshop, Lorne, Australia, 28 February – 3 March 2000. EUM P28, Published by EUMETSAT, D-64295 Darmstadt, 227-234. Available online at http://cimss.ssec.wisc.edu/iwwg/iww5/S5-2_Schmetz-OnTheUse.pdf.
>
> *The references you have found are very fine. Although they are already 20 years old, the conclusions they extracted are completely valid today. It was more or less in the era that the "AMV time scales" were defined and understood. About finding a "peer reviewed reference", general research related of AMVs in general is found in the publications of the "International Winds Wokshops" (http://cimss.ssec.wisc.edu/iwwg/iwwg_meetings.html), but beyond this, much information from these Workshops is not published in peer-reviewed journals (due to costs or the time needed to publish a paper). What I would do here:*

*- Proceedings from the International Winds Workshops are a perfectly valid reference, and the IWWG has the interest to keep them online throughout the time.*

*- Because the main reference you use (Velden, C., D. Stettner, and J. Daniels, 2000) is not available anymore online (only an abstract), I would include the second reference from the 5th International Winds Workshop (Schmetz, J., K. Holmlund, H.P. Roesli, and V. Levizzani, 2000), whose title is atemporal and unrelated to any specific satellite. This can be valid now and in any coming future.*

Thank you for making the point about AMV time scales. We also now add, "Additionally, winds retrieved from tracking clear-sky and cloud motions represent different dynamical features and tend to behave differently. For example, the recommended time interval for tracking cloud motions is 10-15 minutes to capture short cloud lifetimes and rapid intensification/deformation, while the recommended time interval for clear-air motions of 30 minutes is suitable to capture variations in jet streams and other clear-air features (Schmetz et al., 2000)."

*Technical corrections.*

*1. I do not like very much the expression "enhanced wind shear". "Enhanced" reminds me of the word "Improved", which has nothing to do with the wind shear. I would prefer something like "strong wind shear" (throughout all the text). Please update or comment back.*

Agree. Fixed.

*2. You use "rawinsonde" and "radiosonde" in different parts of the text. I assume you are referring with both to the same. Aren't you? This way, use the same word all throughout the text for homogeneity (the one you prefer).*

Yes. We now use *rawinsonde* everywhere.

*3. Line 150: I think you are listing all the LEO satellites and radiometers you are using. So I would remove the text: "including but not limited to".*

Agree. Fixed.

*4. Tables 2 to 5: your options "Bolded and bolded/underlined statistics" do not differentiate well in the text. I suggest to use "Cursive and cursive/underlined statistics" to appreciate better the differences.*

Agree. As an alternate solution, we now replace Tables 2-5 with new Figs. 2-3 that display the information graphically.

*5. In lines 321-322, I think the clouds in "Surface effects may also play a role, as very cold brightness temperatures at or near the polar surface may be misinterpreted as very high cloud tops due to the low temperature contrast between clouds and the surface snow or ice" can actually be located in any height. I would simply change "very high cloud tops" for "cloud tops". Please consider.*

Agree. Fixed.

---

## Author Comment (AC4)

**Response to comments of Anonymous Referee #2 on *AMT* preprint "Exploiting Aeolus Level-2B Winds to Better Characterize Atmospheric Motion Vector Bias and Uncertainty" by Katherine E. Lukens et al. Atmos. Meas. Tech. Discuss., https://doi.org/10.5194/amt-2021-277-RC4, 2021**

Thank you very much for your careful and helpful comments. We have revised our manuscript following your suggestions.

[Below, quotes from your comments are repeated verbatim in *green italic*. Changes in quoted text are indicated as additions and . Comments from author to author are in this color.]

*The paper presents statistics of AMVs vs Aeolus observations from a dataset of collocations. The aim is to evaluate AMVs, with the ultimate goal of improving AMV quality. Aeolus provides an unprecedented dataset in this respect, in particular allowing comparisons of AMVs against other observations in regions where it was previously not possible (ocean, remote land regions).*

*Assessments of the quality of both AMVs and Aeolus data are highly relevant, given they are widely used as input to NWP systems. While there is hence clear merit in producing comparison statistics, I feel a scientific paper requires a clearer interpretation of these statistics than is presently provided. Furthermore, one stated goal of the paper is to guide improvements to AMV quality, but it is not clear to me whether the paper indeed provides new insights into where improvements can be made. It should be possible to address these aspects through a very major revision of the text, particularly in the results and conclusions sections, though some further analysis may also be required to draw firmer conclusions.*

We have followed your suggestions as detailed in our responses below.

*Main general points:*

*1. The paper needs to be clearer on which new insights the study provides and which overall conclusions can be drawn. Presently, the text in section 4 largely textualizes values of statistics given in tables and figures, and it is difficult to grasp what the overall interpretation of these values is and how they link to overall conclusions. One stated goal of the paper is to improve AMV quality. What do we learn about this from the study? For your consideration, Cotton et al (2021) provides a detailed list of features noted in monitoring of AMVs versus NWP, with some of them more clearly attributable to AMVs than others. Could any of these features be investigated with the collocation dataset, hence addressing the stated goal of the paper to aid the development of AMVs?*

We have thoroughly rewritten Section 4 (as well as Section 5), and the key interpretations and conclusions are given in the following quote from the manuscript:

[revised manuscript text omitted]

Additionally, comparisons with Aeolus show another noted feature in monitoring AMVs by Cotton et al. (2020, 2021): a pronounced negative wind speed bias in the tropics for Meteosat-8 is evidenced in this study by large negative MCD and correspond to large SDCD in all regions (**Fig. S1**). This feature is evident in both RAY and MIE comparisons. The fact that comparisons with Aeolus exhibit another known feature hints at the usefulness of Aeolus winds as a standard for comparison to characterize AMVs.

More details are given in the responses below. After carefully considering Cotton et al. (2020, 2021) we decided the direct comparisons possible were limited because all our statistics are for HLOS winds, not vector winds.

*2. What is the basis for stating that "AMVs compare well to Aeolus winds"? What does "well" mean in this context? It appears that the authors compare Aeolus/AMV difference statistics directly to values from AMV/sonde or AMV/NWP comparisons, despite very different uncertainties in the respective comparison datasets or the collocation methods. Uncertainties in Aeolus data are alluded to (incl. biases), but it is not clear how they have been taken into account.*

Thank you for pointing out that the word "well" is not used clearly in this context. In various places we now state the "Overall, GEO and LEO AMVs are found to compare as well with Aeolus RAY and MIE winds as they do to conventional data sources and NWP products."

Additional AMV/NWP statistical comparisons are included in redrawn Figs. 5,7,9 (**new Figs. 7,10,13**). These figures now show profiles of uncertainty estimates with the Aeolus uncertainty removed. A description of this process has been added to the text. We have also added:

"The removal of the Aeolus error estimate results in a smaller SDCD, which still includes AMV random and representativeness errors and collocation error. The SDCD are larger for RAY comparisons than for MIE comparisons in terms of both the original (or total) and adjusted values. Although the Aeolus L2B uncertainty is highly dependent on the time period and processor used to determine the HLOS winds, it is the correct uncertainty estimate for our study."

*3. The statistics presented are affected by collocation/representation error, as well as biased sampling, and my impression is that this may play a considerable role. This aspect should be discussed and, if possible, an attempt at quantifying the magnitude of these aspects should be made.*

We did consider some method to estimate the collocation/representation errors, and we have added a discussion on how this might be done:

"We note that it might be possible to estimate the statistics of the collocation and representativeness errors. The collocation difference may be considered to have three independent components: the error of the AMV winds, the error of the Aeolus winds, and the difference between the truth evaluated for the AMV and the Aeolus winds. We can isolate the first component, the AMV error, if we know the other two components, and we already have estimates of the second component, the Aeolus wind error in the L2B data. The last component is the error due to representativeness and collocation differences. The differences in time and location give rise to the collocation error. The difference in the shapes of the observing volumes gives rise to the representativeness error. If we simulate the AMV and Aeolus observations from a high-quality forecast or analysis or simulation, taken to be the truth, then we can calculate estimates of the combined representativeness and collocation errors. If the truth fields are simply interpolated to the observation locations then the calculated estimates are for the collocation errors alone."

*Specific points:*

*1. Abstract, L26-28: The two sentences appear to contradict each other - on the one hand it is stated that comparisons are consistent with what is known, on the other hand SDCD is over SH is larger than expected.*

The abstract now reads: "In terms of global statistics, QC'd AMVs and QC'd Aeolus HLOSV are highly correlated for both observing modes. …

 Stratified comparisons  with Aeolus HLOSV are consistent with known AMV bias and uncertainty in the tropics, NH extratropics, and in the Arctic, and at mid- to upper-levels in both clear and cloudy scenes."

*2. Abstract L35-39: While I agree with what is stated in this paragraph, this has been recognised for some time (see, for instance, Menzel et al 1996 http://cimss.ssec.wisc.edu/iwwg/iww3/p197-205_Menzel-Improvements.pdf). It seems odd to give such well-established finding such prominence in the present abstract.*

The abstract now reads: "As shown in other comparison studies, the  level of agreement between AMVs and Aeolus wind velocities (HLOSV) varies with the  AMV  type, geographic region, and height of the collocated winds, as well as with the Aeolus observing mode. "

*3. L43/44 ("The survey recommends that radiometry-based …"): The survey considers both radiometry-based AMVs as well as lidar measurements for addressing the requirement of 3d atmospheric winds. To my knowledge, it does not make a recommendation of one versus the other. Please rephrase.*

The text now reads: "The survey  found that radiometry-based atmospheric motion vector (AMV) tracking should be  an important approach to address the priority requirement of 3D winds."

*4. L91/92: Please remove "UTC" in the context of stating overpass times for Aeolus. As stated in the text these are local times, rather than UTC.*

Done. Replaced "UTC" with "LT".

*5. L118-124: Given the high relevance of the Aeolus quality to the present investigation, it would be preferable to give a deeper overview of Aeolus quality assessments, and to refer to peer-reviewed papers on the subject where possible. I am not fully convinced that 3d AIRS AMVs are a suitable reference dataset in this respect. In addition, the statements regarding biases derived from the Santek et al (2021) study appear to be contradictory, with Aeolus showing larger bias against rawinsondes than AIRS AMVs on the one hand, whereas comparisons against ERA5 show similar biases.*

We have replaced lines 118-124 with the following text:

"Recent studies have compared Aeolus winds with various reference wind datasets (e.g., rawinsondes and NWP forecasts). For example, Martin et al. (2021) validated Aeolus HLOS winds against rawinsonde and NWP forecast equivalents for 2018-2019. They found that the estimates of global mean absolute biases and standard deviations of Aeolus based on comparisons with rawinsonde, the ECMWF Integrated Forecasting System (IFS), and the German Weather Service (DWD) forecast model reference datasets are all comparable, with bias magnitudes ranging from 1.8 to 2.3 m s$^{-1}$ for Rayleigh and 1.3 to 1.9 m s$^{-1}$ for Mie and standard deviations ranging from 4.1 to 4.4 m s$^{-1}$ for Rayleigh and 1.9 to 3.0 m s$^{-1}$ for Mie. In addition, the biases vary with latitude and season in a similar way from reference dataset to reference dataset, with the largest differences observed in the tropics and extratropics, particularly during the summer/autumn season. Similarly, Straume et al. (2020) quality assessments showed good correspondence between Aeolus L2B winds and ECMWF model winds for September 2018. Even though Aeolus exhibited random errors that exceeded the mission requirements (4.3 m s$^{-1}$ for Rayleigh) or just met the requirements (2.1 m s$^{-1}$ for Mie), the Aeolus winds still had a positive impact on preliminary NWP experiments. (It should be noted that the results from Martin et al. (2021) and Straume et al. (2020) characterize Aeolus winds before they were reprocessed with the significant M1 wind bias correction applied. The Aeolus bias and error estimates should improve when using the reprocessed winds.)."

In addition, we have reorganized the information presented in the corresponding paragraph to provide a clearer overview of Aeolus quality assessments.

*6. Section 2.2, first 2 paragraphs: Please add which AMV dataset has been used for the various satellites (there are different producers for some of them). I am assuming it is the operational AMV dataset of each satellite operator. I wonder whether the information contained in these paragraphs would be better presented in a table.*

We now add, "AMVs examined in this study (Table 1) are operationally used by the National Oceanic and Atmospheric Administration (NOAA) National Centers for Environmental Prediction (NCEP) and are archived in 6-hour satellite wind (SATWND) BUFR files centered on the analysis times 00, 06, 12, and 18 UTC. All AMVs included in the SATWND files are produced by NESDIS, JMA and EUMETSAT. AMVs derived from sequences of GEO satellite images are observed equatorward of ~60° latitude and are stratified by type, including IR, water vapor cloudy channel (WVcloud), and water vapor clear channel (WVclear) AMVs; visible band AMVs are not used in this study. Polar AMVs (observed at latitudes poleward of 60°) are derived from cloud-tracked IR channels in areas covered by three consecutive LEO satellite images."

Related to this comment, we also changed lines 327-329 to: "The other GEO satellites are not further examined as they exhibit larger SDCD (Meteosat-8 and -11), have a much smaller sample size (Himawari-8,  Meteosat-8 and -11), or are not actively used in NCEP operations (GOES-15 and INSAT 3D)."

Additionally, we have replaced much of the text in the first two paragraphs of Section 2.2 with a **new Table 1** that lists the total collocation counts per satellite as well as the number of observations (and % of total counts) that pass QC for each AMV type and Aeolus observing mode.

*7. Table 1: I find the information condensed in this table very heterogeneous and inconsistent, and I am not convinced that it indeed provides a useful and adequate summary of (all) available monitoring statistics for AMVs. I find the table problematic for a number of reasons:*

*a. While I appreciate the need to condense the information provided, the choice of very broad entries (e.g., all AMVs, all levels, global) appears questionable, given that AMV monitoring statistics vary significantly by season, level, channel, satellite/producer, etc (as apparent from the present paper and many other studies).*

*b. The ranges indicated for some of these statistics are also rather large, and it is difficult to know what these ranges are referring to (presumably some of the variability noted above). At the same time, the very precise numbers given for some datasets (e.g., GOES-16 IR) also do not seem appropriate given the variability with seasons.*

*c. It is not clear why certain references have been selected for some AMV datasets, but not for others (e.g., Cotton et al 2020 and the general NWP SAF monitoring provide monthly statistics for each operational AMV dataset, not only GOES-16 IR). Also, I am sure other papers could be used here to contribute statistics.*

*d. Please note that values given in Cotton et al (2020) are either against the Met Office or the ECMWF system, but not the GFS. The web-address given for the Cotton et al (2020) reference should be updated to https://nwp-saf.eumetsat.int/monitoring/amv/nwpsaf_mo_tr_039.pdf).*

Our reference now points to https://nwp-saf.eumetsat.int/site/monitoring/winds-quality-evaluation/amv/amv-analysis-reports/, the landing page for all the ARs.

*I suggest that the authors critically review the material presented in this table. My impression is that the numbers are primarily used to put the results of the Aeolus/AMV comparisons in broad context, but that these comparisons mostly stay at a rather qualitative level. To stay in line with this qualitative use, the table could also be removed and replaced with a simple statement of typical values found in collocation statistics.*

We have removed Table 1 and replaced it with the following text: "AMV performance metrics vary significantly by season, level, channel, satellite/producer, etc. (e.g., Santek et al., 2019; Daniels et al., 2018; Cotton et al., 2020; Key et al., 2016; Le Marshall et al., 2008). For example, typical values of AMV wind speed bias acquired from seven different data producers and verified against rawinsonde winds can range from -1.8 m s$^{-1}$ to 0.3 m s$^{-1}$, and wind speed uncertainty represented by standard deviation can range from 4 to 6.5 m s$^{-1}$, with higher vector wind root mean square errors of 6-9 m s$^{-1}$. Even for a single satellite, e.g., GOES-16 or Aqua, speed bias and uncertainty can vary geographically as well as vertically."

*8. Table 1: Which QI has been used to quality-control the AMVs (forecast-dependent or independent) for the studies shown? The choice of QI can have a significant impact on monitoring statistics.*

The text now states that "For all AMVs, a forecast-independent quality indicator (QI) of at least 80% is used to filter and retain the high-quality data."

*9. L187-192 ("Since it takes approximately 92 min... closest in the vertical to the AMV."): I struggle to understand these sentences. Are the authors saying that if multiple Aeolus winds fulfil*

*the collocation criterion then the Aeolus profile closest is space is used, and within that profile the observation closest in pressure?*

Yes, as stated at L177: "Then, if multiple Aeolus observations still meet all collocation criteria, the observation closest in pressure to the AMV observation is kept for analysis." After this sentence we add "There is no need to consider closeness in time given the collocation criteria and the Aeolus orbit." And eliminate the text in L187-192.

*10. L187-192: Are Mie/cloudy and Rayleigh/clear winds collocated separately here or are they treated together? Ie, could the same AMV be collocated once with a Mie/cloudy Aeolus wind and once with a Rayleigh/clear wind?*

At the beginning of this section we add, "AMV collocation datasets are prepared separately for RAY and MIE winds. (A single AMV might appear in both data sets.)"

*11. L194-203: I note that the text does not mention an outlier removal (ie removal of collocations that show particularly large deviations). Please confirm that no outlier removal has indeed been applied. I note the absence of egregious outliers in Figures 4 and 6, hence the question.*

At the end of this paragraph we add, "No explicit outlier QC is applied and since there are no extreme outliers (seen below in Figs. 6 and 9), the QC that is applied is sufficient to eliminate them."

*12. L201: Which QI has been used for quality control in this study?*

See our response to point 8.

*13. Fig. 2: Please clarify what grid cells have been used in this plot and whether they are of equal area size. The caption states that each grid cell is 1.25° or 140 km, but the former would lead to progressively smaller cells at high latitudes and is incompatible with the 140 km.*

The figure captions now read, "within a grid cell at 1.25° (approximately 140 km in the N-S direction) horizontal resolution".

*14. Tables 2-5: Given the considerable variations shown by channel, wind type, level in Fig. 5 (and other Figures), how useful are the statistics given in these tables? Also, it is very difficult to grasp the information conveyed in this way - would replacing the table with a graphical display help?*

This is an important point. To be clear the first paragraph of Section 4 now includes: "In agreement with previous studies, our results confirm that the level of agreement between AMVs and Aeolus winds varies per combination of conditions including the observing scene type (clear vs. cloudy) coupled with AMV type, geographic region, and height of the observable. Moreover, the findings highlight the value of using Aeolus MIE winds as a comparison standard to characterize AMVs. For context, we begin with summary statistics for samples that include all conditions."

In addition, Tables 2-5 have been replaced with **new Figs. 2 and 3** that display the same statistical information presented in the tables. The statistics are easily compared for each satellite in each geographic region and clearly show that RAY comparisons tend to have larger SDCD

than MIE, and MCD and SDCD in the SH are generally larger than the other regions. The results are summarized in the text.

*15. L274/275 ("Overall, GEO AMVs correspond very well with RAY and MIE winds..."): See general point 2 above.*

The text now reads: " The main points from the summary collocation statistics of RAY and MIE winds with AMVs are the following: …"

*16. Figure 4: the text on the plot is very small and hence difficult to read (ie axis labels, and summary statistics).*

Figures 4, 6, and 8 (**new Figs. 6, 9, and 12**) have been redrawn.

*17. 4.1.1: A common thread throughout this sub-section seems to be the finding that WV clear AMVs compare better with the Rayleigh/clear winds than the cloudy IR or WV winds (stated multiple times). I think some critical discussion of this finding would be useful. The fact that cloudy AMVs were found in a region where Aeolus indicates a clear scene suggests that either the AMV height assignment is erroneous or that collocation/representation errors are likely to be larger (as the Aeolus wind must originate from a different area than the AMV). So by design these statistics for cloudy AMVs are expected to be less favourable than the ones for clear AMVs. Without further analysis the statistics will give little insight in the relative quality of clear-sky AMVs vs cloudy AMVs in general.*

We have expanded our discussion of this finding. The manuscript now reads:

"Of the three AMV types, the best match is for WVclear AMVs, with the comparisons exhibiting the smallest SDCD values in each geographic region, that in turn are comparable to known wind speed SD and RMS of all GEO AMVs relative to rawinsonde winds (Santek et al., 2019). This is expected since WVclear AMVs and Aeolus RAY winds are most probably sampling similar clear-sky scenes, and clear scenes are more homogeneous over time and space scales, which in turn implies smaller collocation differences. Ideally, one would expect samples large enough to provide statistically significant collocation differences between RAY winds and WVclear AMVs only; as it turns out, collocation differences are also statistically significant for IR AMVs (see Fig. 7). In these cases cloudy AMVs are collocated with Aeolus RAY winds that represent clear scenes, and since they do not observe the same type of scene, Aeolus and/or AMV representativeness errors are most probably larger (hereafter we refer to this as the cloudy/clear sampling effect).

"To better isolate the AMV error, the Aeolus error estimate is removed from the SDCD at each level, resulting in mean profiles of adjusted SDCD (long dashed lines in Figs. 7b, 7e, and 7h) that include AMV errors and collocation/representativeness errors. Overall, the adjusted SDCD for all AMV types exhibit similar magnitudes and distributions in each geographic region throughout the vertical. WVclear comparisons have slightly smaller adjusted SDCD at upper levels, suggesting that sampling differences may play a role in the higher accuracy observed for WVclear AMVs, given that WVclear representativeness errors are likely small due to Aeolus RAY and WVclear AMVs observing similar scenes. Aeolus RAY uncertainty is larger in the presence of clouds and appears to have a considerable impact on the corresponding SDCD, as the reductions in IR and WVcloud SDCD ($\sim$1 m s$^{-1}$) are larger than for WVclear (0.5 m s$^{-1}$).

"Regarding GOES-16 RAY comparisons, sampling differences may play a role in the higher correlation between Aeolus RAY winds and WVclear AMVs, since they both represent similar clear-sky scenes. This is especially true in the tropics and NH extratropics where MCD are small and SDCD are comparable to AMV error values compared with high-quality rawinsonde winds. It is likely that collocation errors play a larger role in the RAY SDCD for IR and WVcloud AMVs due to the cloudy/clear sampling effect, where clear-sky Aeolus winds are collocated with cloudy AMVs and thereby observe different scenes, yielding larger errors."

*18. 4.1.2: Related to the above, I note that in this section clear-sky AMVs are excluded from comparisons with Mie/cloudy Aeolus winds, with the argument that clear-sky AMVs measure wind in clear scenes only. The choice is inconsistent with the choice made in 4.1.1, where comparisons of cloudy AMVs vs Rayleigh/clear Aeolus winds were included. Could the authors elaborate on the reasons for these two different choices?*

We now add, "To increase the size of our collocation data set, we compared all types of GOES-16 AMVs to both Rayleigh-clear and Mie-cloudy winds. In addition, we do not show results from WVclear AMV collocations with Mie-cloudy winds as correlations for this category of collocations are poor and the sample size is very small (see Table 1), and this result may be unreliable. With a larger data set it might be possible to compare Rayleigh-clear and Mie-cloudy winds to clear and cloudy AMVs only, respectively. Additionally, winds retrieved from tracking clear-sky and cloud motions represent different dynamical features and tend to behave differently. For example, the recommended time interval for tracking cloud motions is 10-15 minutes to capture short cloud lifetimes and rapid intensification/deformation, while the recommended time interval for clear-air motions of 30 minutes is suitable to capture variations in jet streams and other clear-air features (Schmetz et al., 2000)."

*19. 4.1.2: Similar to the point above, the effect of the sampling imposed by looking at Mie/cloudy vs cloudy AMV collocations should be discussed here. By design this is a sample where Aeolus and AMVs agree in terms of a cloud being present at a particular altitude. So this sample of AMVs would be expected to have smaller height assignment errors (as the height assignment has effectively been quality-controlled by Aeolus), and representation errors are likely to be smaller (as AMVs and Aeolus are more likely sampling similar areas). This will contribute to favourable comparison statistics. Of course, the smaller random error in the Mie/cloudy wind is another reason for smaller SDSCs compared to values shown in 4.1.1. Based on Aeolus uncertainty estimates, is it possible to quantify which aspect is the dominant factor?*

We now add, "MIE SDCD are considerably smaller than those for RAY comparisons, and this is attributed to the general higher accuracy of Aeolus MIE wind retrievals. Another possible reason is that MIE comparisons might generally have smaller collocation errors: because collocated Aeolus MIE winds and IR and WVcloud AMVs are by definition more likely sampling similar cloudy scenes at similar altitudes, we expect the Aeolus and AMV random and representativeness errors to be small (hereafter the cloudy/cloudy sampling effect).

"…Aeolus MIE winds show great potential value as a comparison standard to characterize cloudy AMVs. MIE comparisons generally exhibit smaller biases and uncertainties compared to RAY, reflecting the higher accuracy of MIE winds and cloudy AMVs in cloudy scenes as well as larger collocation errors for RAY winds in cloudy scenes. This is attributed to a combination of smaller Aeolus MIE uncertainties and smaller collocation/representativeness errors due to the cloudy/cloudy sampling effect, that is, the fact that both Aeolus and AMV winds are, by

definition, sampling similar cloudy scenes at similar altitudes. The contribution of Aeolus MIE uncertainty to the overall SDCD is small; in fact, removal of Aeolus uncertainties further reduces the small MIE SDCD without much change to its vertical distribution, suggesting that for MIE comparisons, the dominant factors contributing to the total error consist of AMV random errors and representativeness/collocation errors."

*20. L450-453: The relatively large systematic differences over the SH extra-tropics appear to be attributed to AMVs, as the authors suspect height assignment errors. Are there any reasons to believe that Aeolus winds could be in error in this particular region?*

To address this comment, we now include in **new Figs. 7, 10, and 13** profiles of the mean AMV wind speed (not projected onto HLOS) from which one can infer vertical wind shear by the gradient of the AMV wind speed with respect to pressure. In addition, we include figures of the decomposed AMV error and corresponding Aeolus error with respect to AMV wind speed (**new Figs. 8, 11, and 14**). The manuscript now states:

"MCD are largest in the SH extratropics and are statistically significant throughout the vertical, ranging from -1.0 m s$^{-1}$ at low levels to < -3.0 m s$^{-1}$ above 300 hPa. Strong wind shear corresponding to an intensified jet is inferred at upper levels (Fig. 10g). The larger MCD aloft are associated with increases in adjusted SDCD with height, which are on the order of 4-5 m s$^{-1}$. Moreover, the large MCD represent over 8.5% of the corresponding HLOSV at upper levels and could be attributed to larger AMV height assignment errors corresponding to stronger storm tracks in winter. This is exemplified in Fig. 11e where Aeolus MIE errors are shown to be small (2 m s$^{-1}$) and remain relatively constant with increasing AMV wind speed, while the adjusted SDCD are larger and increase with AMV wind speeds > 40 m s$^{-1}$. The results imply that the large systematic differences in MCD at upper levels in the SH extratropics are most probably attributed to a combination of larger AMV errors in combination with strong wind shear."

*21. L454-455: The biases exceeding -3 m/s over the SH extra-tropics are not small, and they are not in line with the ranges given in Table 1. This seems to be acknowledged later in the same paragraph (L459-460), but the sentences in question expresses the opposite.*

Thanks for catching that. The text now reads, "Statistics of AMV minus MIE collocation differences are generally consistent, albeit with some notable exceptions, with those for AMV comparisons with high quality rawinsonde winds."

*22. L536-538 ("Overall, GEO and LEO AMVs are found to correspond very well with Aeolus RAY and MIE winds... range of known biases and uncertainties of AMVs"): See general point 2 above.*

The text now reads, "Overall, GEO and LEO AMVs are found to compare as well  with Aeolus RAY and MIE winds as they do to conventional data sources and NWP products."

*23. L550 ("GOES-16 AMVs are found to compare well with RAY and MIE winds"): As above, see general point 2.*

The text now reads: "The main findings from comparing GOES-16 AMVs  with RAY and MIE winds are the following."

*24. L552-553 ("WVclear AMVs perform best..."): See earlier point 17. This is likely at least partially due to biased sampling, and without further analysis it would be inappropriate to conclude that WVclear AMVs are more accurate than WVcloudy AMVs. This should be clearly addressed when interpreting the results. A similar comment applies to abstract L29/31.*

See our response to point 17.

*25. L570 ("... Aeolus could be used as a standard for the comparative assessment of AMVs pending additional bias corrections to the Aeolus L2B winds"): I am not sure what the authors are saying here. Are the results presented not reliable, as additional bias correction for Aeolus winds is required? Or do the authors think that their results suggest that additional bias correction is needed for Aeolus? I don't think there is sufficient evidence for either statement, so I am puzzled what is meant here. A similar comment applies to abstract L25/26.*

Our results are reliable. We now add: "The Aeolus project has done much to eliminate errors of all types, but some improvements are expected. For example, some of the bias corrections currently applied depend on ECMWF forecasts and the analysis of Liu et al. (2021, TLSBC) demonstrates that additional bias correction for Aeolus are possible, and that such corrections can improve NWP analysis and forecast results (Garrett et al., 2021, Impacts)."

**Supplemental Figure**

**Figure S1**: As in manuscript Fig. 1 but showing Meteosat-8 vertical profiles of AMVs collocated with Aeolus RAY (top) and MIE winds (bottom) in the tropics.

[Figure]

---

## Referee Report (RR1)

Dear authors,

The large majority of the items I remarked in my previous review has been corrected in a satisfactory way. However I still see a few of them (shown with ➔ below) which still need a second round of review, especially points 1c) and 7), which are the more important ones.

Here I have not taken care of the elements for review by the other Reviewer. But they are very important and they should also be taken into account. This paper should not be published until the other Reviewer is happy with corresponding answers.

Best regards

*Specific comments*

*1a. Please include a description, including a formula, etc*
➔ I am happy with the inclusion of Appendix A, which makes statistics more clear. However I am still missing some information there: the definition in the Appendix of HLOSV as "Horizontal line-of-sight velocity", and the fact that you are only able to calculate statistics in this specific wind component. Please add this info there.

Answers to point 1b: OK

*1c. The collocation criteria used between AMVs and Aeolus winds, etc*
➔ I am happy with the explanation, which is very clear. Although with what you say now I could find pressure differences of 300 hPa at 1000 hPa levels too wide.
➔ I understand that you cannot change these values now to rerun all statistics, and at the same time these values optimize your results.
➔ With all this, you could add some sentence at the end of this paragraph saying that "these pressure differences might be a bit wide at the lowest levels, but they have optimized the size of the validation sample".

Answer to point 2a: OK

Answer to point 2b: OK,
➔ Although I still see two "AMV channel type" in Figures 1 and 6 to be changed.

Answers to points 3, 4a, 4b: OK

Answer to point 4c: OK,
➔ Although you should change "blocklisted" word for "blacklisted" (twice in the text!).

Answer to points 5a, 5b, 6: OK

Answer to point 7: I am not so happy with this way of doing. You know your datasets and how much the division of data in two groups is possible or not (Clear air AMVs with Rayleigh clear winds on one side; Cloudy AMVs and Mie cloudy winds on the other side). You explain that your sample size is small and you cannot do this division. But in the second sentence

"Additionally, winds retrieved from tracking clear-sky and cloud motions represent different dynamical features and tend to behave differently"
you clearly say that both wind types behave differently, and so mixing all GOES AMVs together should not be done.
➔ If you keep everything this way, you should try a way to put together both sentences without arguing with each other. Because otherwise readers of your paper would not be happy.

*Technical corrections.*
OK with all of them; especially the change of Tables 2-5 to Figures 2-3, much more clear.

This is a comment by Anonymous Referee #1, on the questions defined by the other Referee in the first round of review of this paper.

He/she has not answered to the review of this paper in this second round, which is a surprise to me due to the important review process which was requested by him/her in the first round.

I am checking if all his/her questions have been answered satisfactorily. Please reconsider those elements defined in ==yellow== in a small editorial further review.
* * *
Main comments
 - "I feel a scientific paper requires a clearer interpretation of these statistics than is presently provided"
➔ The paper has been reviewed with detail, and I consider the information provided in the latest version is clear enough, and not prone to doubts. It can be published taking into account the few elements remarked in yellow below.

 - "One stated goal of the paper is to guide improvements to AMV quality, but it is not clear to me whether the paper indeed provides new insights into where improvements can be made. It should be possible to address these aspects through a very major revision of the text, particularly in the results and conclusions sections, though some further analysis may also be required to draw firmer conclusions"
➔ Please consider what is said in "Main general point 2"
* * *
Main general points:

1. About "The paper needs to be clearer on which new insights the study provides and which overall conclusions can be drawn; One stated goal is to improve AMV quality. What do we learn about this from the study? Could any of these features be investigated with the collocation dataset, hence addressing the stated goal of the paper to aid the development of AMVs?"
➔ As said above, The paper has been reviewed with detail, and I consider the information provided in the latest version is clear enough, and not prone to doubts. The other reviewer is asking for several additions about "how to improve AMV quality and aid the development of AMVs", but I consider this can be outside of the scope of this paper, and additionally, this paper needs to be published soon so that corresponding findings can be available to everybody.
➔ My suggestion here could be: if you have the possibility and the interest, think of writing another paper later which can based on these ideas: "Could any of these features be investigated with the collocation dataset, hence addressing the stated goal of the paper to aid the development of AMVs? How should affected AMVs change to improve their quality using Aeolus winds as reference?"

2. About "What is the basis for stating that "AMVs compare well to Aeolus winds"? What does "well" mean in this context? It appears that the authors compare Aeolus/AMV difference statistics directly to values from AMV/sonde or AMV/NWP comparisons, despite very different uncertainties in the respective comparison datasets or the collocation methods."

➔ All the paper has been rewritten, and this ambiguous comment "AMVs compare well to Aeolus winds" has been fully removed from the text, specifying clearly how AMVs/Aeolus winds match and differ, and what can be the cause to these differences. With this, I think this point has been solved.

3. About "The statistics presented are affected by collocation/representation error, as well as biased sampling, and my impression is that this may play a considerable role. This aspect should be discussed and, if possible, an attempt at quantifying the magnitude of these aspects should be made"

➔ This question has been thoroughly answered in lines 212-270.
* * *
Specific points:
1. The sentences have been rewritten, and I do not see contradictions anymore.
2. The abstract has been reduced, removing the mentioned lines from the text.
3. The text has been rewritten as requested.
4. UTC changed for LT as requested.

5. The text has been changed and explained in more detailed, such as requested by the reviewer. However, I see in the text two "blocklisted" in lines 86 and 121 that should be changed to "blacklisted".

6. The text has been changed as requested and presented in a table with more information, such as suggested. However, I see here again that you consider again "water vapor cloudy channel (WVcloud), and water vapor clear channel (WVclear) AMVs" as if coming from different satellite channels. As I already said in my first review, WVcloud and WVclear can be calculated using the same WV channel, tracking different features (clouds or moisture). Rephrase again here and in any other part of the text where this occurs again as: "water vapor cloud AMVs (WVcloud), and water vapor clear air AMVs (WVclear)"

7. The content has been changed such as requested, making clear the big variability of results considering different processing centres and circumstances. I think the information is successfully provided in a more qualitative way, such as requested.
8. Table 1 has been removed, and so this comment does not apply anymore.
9. The text is more clearly presented and easier to understand.
10. The question has been answered in the text.
11. The question has been answered in the text.

12. Please specify also in line 145 that you are using the "forecast-independent QI".
13. Please include the clarification "1.25° (approximately 140 km in the N-S direction)" in both Figures 4 and 5.

14. Tables 2-5 have been changed to Figures 2-3, which are clearer and easier to read.
15. Answered in "Main general point 2"
16. Text in the figure has been made bigger, and now it can be seen better.

17/18/19. These three comments define again an important question I made in the previous review round: it could have been more useful to compare Clear air AMVs with Rayleigh/clear air Aeolus winds only, and Cloudy AMVs with Mie/cloudy Aeolus winds only. The study would have been more helpful.

You already gave your reasons to do what you did, and now it would be very late to change all this. But at least you could define with some more detail why you did this way (why you could not do as a I say above).

In your text, the only explanation I find is in lines 376-380, which say "To increase the size of our collocation dataset we compared all types…" and "With a larger data set it might be possible to compare Rayleigh-clear and Mie-cloudy winds to clear and cloudy AMVs only, respectively". I find this explanation weak, considering the implications this has had in the results.

Beyond this, all the paper has been rewritten following "Main general point 2", specifying clearly how AMVs/Aeolus winds match and differ, and what can be the cause to these differences.

20. I agree with the conclussion provided.
21. I agree with the conclussion provided.
22. Answered in "Main general point 2"
23. Answered in "Main general point 2"
24. Answered in "Main general point 2"
25. I agree with the conclussion related to this provided in lines 672-680.

---

## Author Response (AR3)

**Response to comments of Anonymous Referee #1 on *AMT* preprint "Exploiting Aeolus Level-2B Winds to Better Characterize Atmospheric Motion Vector Bias and Uncertainty" by Katherine E. Lukens et al. Atmos. Meas. Tech. Discuss., https://doi.org/10.5194/amt-2021-277-RC1, https://doi.org/10.5194/amt-2021-277-RC2, https://doi.org/10.5194/amt-2021-277-RC3, 2021**

Thank you very much again for your careful and helpful comments. We are grateful for your thorough review of our manuscript, its revisions, and our responses to all referee comments. The active dialogue to clarify these comments was particularly useful in improving the article. We have revised our manuscript following your suggestions.

[Below, quotes from your comments are repeated verbatim in *green italic*. Changes in quoted text are indicated as additions and .]

*This is a comment by Anonymous Referee #1, on the questions defined by the other Referee in the first round of review of this paper.*

*He/she has not answered to the review of this paper in this second round, which is a surprise to me due to the important review process which was requested by him/her in the first round.*

*I am checking if all his/her questions have been answered satisfactorily. Please reconsider those elements defined in yellow in a small editorial further review.*

*---*

*Main comments*
 - *"I feel a scientific paper requires a clearer interpretation of these statistics than is presently provided"*
 → *The paper has been reviewed with detail, and I consider the information provided in the latest version is clear enough, and not prone to doubts. It can be published taking into account the few elements remarked in yellow below.*

Thank you. We have considered all points and have revised our manuscript accordingly.

 - *"One stated goal of the paper is to guide improvements to AMV quality, but it is not clear to me whether the paper indeed provides new insights into where improvements can be made. It should be possible to address these aspects through a very major revision of the text, particularly in the results and conclusions sections, though some further analysis may also be required to draw firmer conclusions"*
 → *Please consider what is said in "Main general point 2"*

*---*

*Main general points:*

1. *About "The paper needs to be clearer on which new insights the study provides and which*

*overall conclusions can be drawn; One stated goal is to improve AMV quality. What do we learn about this from the study? Could any of these features be investigated with the collocation dataset, hence addressing the stated goal of the paper to aid the development of AMVs?"*

*→ As said above, The paper has been reviewed with detail, and I consider the information provided in the latest version is clear enough, and not prone to doubts. The other reviewer is asking for several additions about "how to improve AMV quality and aid the development of AMVs", but I consider this can be outside of the scope of this paper, and additionally, this paper needs to be published soon so that corresponding findings can be available to everybody.*

*→ My suggestion here could be: if you have the possibility and the interest, think of writing another paper later which can based on these ideas: "Could any of these features be investigated with the collocation dataset, hence addressing the stated goal of the paper to aid the development of AMVs? How should affected AMVs change to improve their quality using Aeolus winds as reference?"*

Thank you for this suggestion. We will certainly consider writing an additional paper addressing these topics.

*2. About "What is the basis for stating that "AMVs compare well to Aeolus winds"? What does "well" mean in this context? It appears that the authors compare Aeolus/AMV difference statistics directly to values from AMV/sonde or AMV/NWP comparisons, despite very different uncertainties in the respective comparison datasets or the collocation methods."*

*→ All the paper has been rewritten, and this ambiguous comment "AMVs compare well to Aeolus winds" has been fully removed from the text, specifying clearly how AMVs/Aeolus winds match and differ, and what can be the cause to these differences. With this, I think this point has been solved.*

*3. About "The statistics presented are affected by collocation/representation error, as well as biased sampling, and my impression is that this may play a considerable role. This aspect should be discussed and, if possible, an attempt at quantifying the magnitude of these aspects should be made"*

*→ This question has been thoroughly answered in lines 212-270.*

*---*

*Specific points:*
*1. The sentences have been rewritten, and I do not see contradictions anymore.*
*2. The abstract has been reduced, removing the mentioned lines from the text.*
*3. The text has been rewritten as requested.*
*4. UTC changed for LT as requested.*
*5. The text has been changed and explained in more detailed, such as requested by the reviewer. However, I see in the text two "blocklisted" in lines 86 and 121 that should be changed to "blacklisted".*

We have added a footnote about this terminology in the manuscript at line 86:

"The meaning of the term "blocklist" is identical to the term "blacklist"; however,

"blacklist" has racist connotations (Conger, 2021). The term "blocklist" is intentionally used in an effort to support the usage of more neutral computing terminology in scientific research; in fact, the Aeolus project has already adopted this new terminology and refers to the list of dates when Aeolus data should be excluded as "blocklisted" dates."

6. *The text has been changed as requested and presented in a table with more information, such as suggested. However, I see here again that you consider again "water vapor cloudy channel (WVcloud), and water vapor clear channel (WVclear) AMVs" as if coming from different satellite channels. As I already said in my first review, WVcloud and WVclear can be calculated using the same WV channel, tracking different features (clouds or moisture). Rephrase again here and in any other part of the text where this occurs again as: "water vapor cloud AMVs (WVcloud), and water vapor clear air AMVs (WVclear)"*

Thanks for catching that. All instances of this phrase that appeared in the text have been rephrased accordingly.

7. *The content has been changed such as requested, making clear the big variability of results considering different processing centres and circumstances. I think the information is successfully provided in a more qualitative way, such as requested.*
8. *Table 1 has been removed, and so this comment does not apply anymore.*
9. *The text is more clearly presented and easier to understand.*
10. *The question has been answered in the text.*
11. *The question has been answered in the text.*
12. *Please specify also in line 145 that you are using the "forecast-independent QI".*

Done.

13. *Please include the clarification "1.25° (approximately 140 km in the N-S direction)" in both Figures 4 and 5.*

Done.

14. *Tables 2-5 have been changed to Figures 2-3, which are clearer and easier to read.*
15. *Answered in "Main general point 2"*
16. *Text in the figure has been made bigger, and now it can be seen better.*
*17/18/19. These three comments define again an important question I made in the previous review round: it could have been more useful to compare Clear air AMVs with Rayleigh/clear air Aeolus winds only, and Cloudy AMVs with Mie/cloudy Aeolus winds only. The study would have been more helpful.*

*You already gave your reasons to do what you did, and now it would be very late to change all this. But at least you could define with some more detail why you did this way (why you could not do as a I say above).*

*In your text, the only explanation I find is in lines 376-380, which say "To increase the size of our collocation dataset we compared all types…" and "With a larger dataset it might be possible to compare Rayleigh-clear and Mie-cloudy winds to clear and cloudy AMVs only, respectively". I find this explanation weak, considering the implications this has had in the results.*

We now include the following text in Section 5 of the manuscript:

"Future studies should use larger datasets like those the authors are preparing to compare clear-scene AMVs with Aeolus Rayleigh-clear winds only, and cloudy-scene AMVs with Aeolus Mie-cloudy winds only. Such studies are anticipated to yield additional insights into the seasonal performance of AMV characteristics representing different dynamical features in clear and cloudy scenes and how this might be accounted for or improved upon in AMV algorithms. Moreover, the robustness of dynamical features identified in AMV monitoring could be further validated following this approach. In addition, Aeolus Mie-cloudy comparisons using larger datasets are expected to have a significant impact towards improving our understanding and characterization of AMV quality in cloudy scenes, given the cloudy/cloudy sampling effect and the small contribution of Aeolus Mie-cloudy error to the total SDCD throughout the vertical in all geographic regions, implying that the corresponding adjusted SDCD better depicts true AMV uncertainty. This is especially critical where AMV height assignment errors are likely large but Aeolus Mie-cloudy errors are small and remain relatively constant with respect to height and AMV wind speed, e.g., in layers of strong vertical wind shear and in the SH."

*Beyond this, all the paper has been rewritten following "Main general point 2", specifying clearly how AMVs/Aeolus winds match and differ, and what can be the cause to these differences.*

20. *I agree with the conclussion provided.*
21. *I agree with the conclussion provided.*
22. *Answered in "Main general point 2"*
23. *Answered in "Main general point 2"*
24. *Answered in "Main general point 2"*
25. *I agree with the conclussion related to this provided in lines 672-680.*